# Orchestrating Spatial Semantics via a Zone-Graph Paradigm for Intricate Indoor Scene Generation

Meisheng Zhang [1 2]   Shizhao Sun [2]   Yang Zhao [3 2]   Ziyuan Liu [1 2]   Zhijun Gao [1]   Jiang Bian [2]

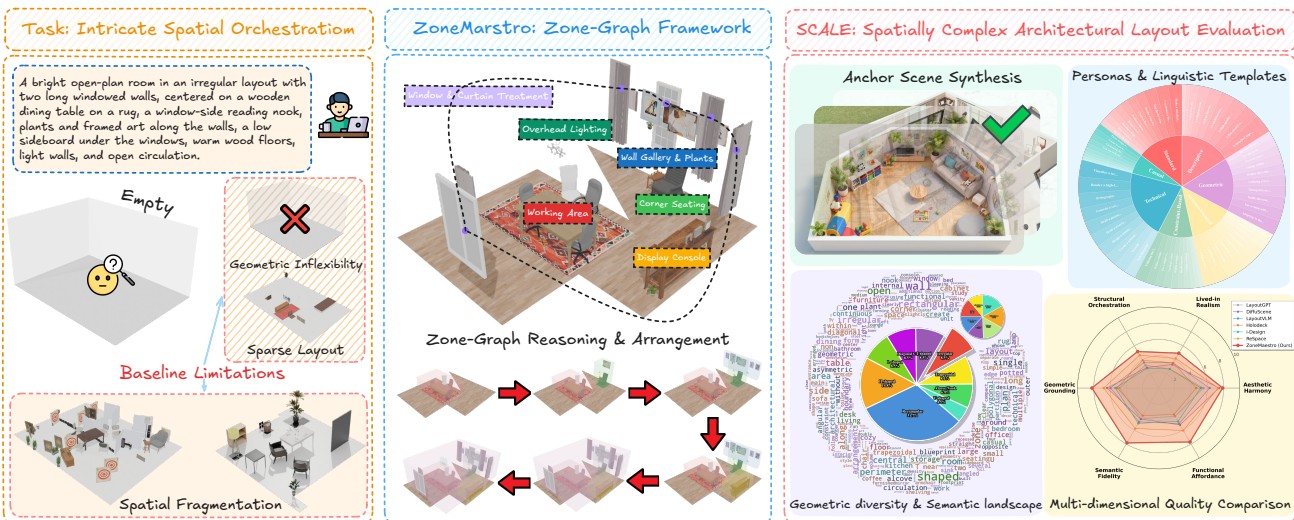

*Figure 1.* **Overview of ZoneMaestro.** Left: Existing methods suffer from geometric inflexibility and spatial fragmentation. Middle: Our Zone-Graph framework decomposes the task into compositional reasoning and spatial orchestration. Right: ZoneMaestro achieves superior coherence and density on the SCALE benchmark.

## Abstract

Autonomous 3D indoor scene synthesis breaks down in non-convex rooms with tightly coupled spatial constraints. Data-driven generators lack topological priors for long-horizon planning, while iterative agents fragment semantics and become geometrically brittle. We present **ZoneMaestro**, a unified framework that shifts the paradigm from object-centric synthesis to Zone-Graph Orchestration. By internalizing a novel zone-based logic, ZoneMaestro translates high-level semantic intent into functional zones and topological constraints, enabling robust adaptation to diverse architectural forms. To support this, we construct *Zone-Scene-10K*, a large-scale dataset enriched with explicit Zone-Graph annotations. We further introduce an *Alternating Alignment Strat-*

*egy* that cycles between reasoning internalization and Zone-Aware Group Relative Policy Optimization (*Z-GRPO*), effectively reconciling the tension between semantic richness and geometric validity without relying on external physics engines. To rigorously evaluate spatial intelligence beyond convex primitives, we formally define the task of **Intricate Spatial Orchestration** and release SCALE, a stress-test benchmark for irregular indoor scenarios with complex, dense spatial relations. Extensive experiments demonstrate that ZoneMaestro resolves the density-safety dichotomy, significantly outperforming state-of-the-art baselines in both structural coherence and intent adherence.

## 1. Introduction

Synthesizing 3D indoor environments with semantic and structural fidelity is essential for advancing embodied artificial intelligence, including embodied rearrangement (Wu et al., 2023; Ding et al., 2024), spatial grounding (Chen et al., 2024; Jatavallabhula et al., 2023), scene graph prediction (Gu et al., 2024), and long-horizon simulation (Puig et al., 2023; Li et al., 2024). While language-driven syn-

Work was done during an internship at Microsoft Research Asia. [1]Peking University, Beijing, China [2]Microsoft Research Asia, Beijing, China [3]Shanghai Jiao Tong University, Shanghai, China. Correspondence to: Shizhao Sun <shizsu@microsoft.com>.

*Proceedings of the 43rd International Conference on Machine Learning*, Seoul, South Korea. PMLR 306, 2026. Copyright 2026 by the author(s).

thesis has mastered simple canonical layouts, its capability diminishes significantly in the regime of Intricate Spatial Orchestration. This domain demands the generation of scenes characterized by dense spatial relationships within irregular, non-convex boundaries. Unlike idealized box-shaped settings, realistic scenarios require managing entangled spatial dependencies rather than mere object quantities, a complexity essential for bridging language instructions with physical spatial realities.

Current approaches largely struggle to enforce structural priors in such complex environments. Methods relying on explicit intermediate plans face a grounding gap between abstract relations and metric coordinates, often correcting violations myopically without restoring global structure (Feng et al., 2023; Sun et al., 2025a; Wong et al., 2025). Agentic systems rely on reactive simulation feedback which is often rigid, and multi-step refinement can accumulate deviations that weaken instruction fidelity (Yang et al., 2025b; 2024c; Çelen et al., 2024; Hu et al., 2025). Terminal reinforcement learning alignment favors constraint-satisfying shortcuts, locking in early decisions and degrading global coherence (Bucher & Armeni, 2025; Yang et al., 2025c; Ran et al., 2025; Pan & Liu, 2025). Other data-driven generators remain brittle when non-convex boundaries interact with dense inter-object coupling, leading to compounded placement errors (Paschalidou et al., 2021; Tang et al., 2024; Yang et al., 2024a;b).

We propose that handling spatial complexity requires a unified **Zone-Graph Paradigm**. We introduce ZoneMaestro, an LLM-driven framework that reformulates generation through Zone-Graph Reasoning in Figure 1. Unlike methods treating space as a continuous vacuum, ZoneMaestro perceives architectural complexity as a topological graph of functional containers. This *cognitive reconfiguration* enables two capabilities. *Geometric Adaptation* lets zones deform to occupy non-convex recesses. *Semantic Encapsulation* isolates high-density dependencies to prevent the semantic drift that plagues long-horizon autoregression. By internalizing this logic, the model transitions from merely placing objects to curating spatial narratives. To support this, we construct *Zone-Scene-10K*, enriching InternScenes (Zhong et al., 2025) with explicit Zone-Graph reasoning annotations. We further devise an *Alternating Alignment Strategy* cycling between Zone Reasoning Internalization and Zone-Aware Group Relative Policy Optimization (Z-GRPO). By iteratively cycling between intrinsic reward optimization and reasoning consolidation, we prevent the semantic degradation typical of pure RL. This effectively reconciles diverse spatial arrangement with rigorous physical compliance.

Formalizing the distinct capabilities required to navigate this regime, we define the task of **Intricate Spatial Orchestration**. This formulation transcends the simple population of convex hulls, demanding that systems strictly satisfy the topological conflicts between high-density semantic intent and valid geometric execution. However, existing protocols overlook the critical failures inherent to non-convex regimes (Lin & Mu, 2024; Hao et al., 2025; Tam et al., 2026). To drive research beyond current idealized settings, we release **SCALE** (**S**patially **C**omplex **A**rchitectural **L**ayout **E**valuation). By isolating the failure modes of standard baselines, especially their inability to maintain structural coherence within non-convex boundaries, SCALE establishes the first rigorous benchmark for measuring spatial intelligence in realistic, architecturally diverse environments.

In summary, our contributions are as follows:

- We introduce ZoneMaestro, which reformulates layout synthesis via the **Zone-Graph Paradigm**. This approach enables semantic encapsulation and geometric adaptation to non-convex boundaries, overcoming the topological myopia of linear baselines.
- We devise **Alternating Spatial Alignment**, cycling between Zone Reasoning Internalization and Zone-Aware GRPO. This reconciles spatial diversity with physical rigor, eliminating geometric noise without semantic degradation.
- We formalize the task of Intricate Spatial Orchestration and release **SCALE**, Spatially Complex Architectural Layout Evaluation. This benchmark isolates failure modes in non-convex topologies, demonstrating our superior structural coherence over existing paradigms.

**Conflict of Interest Disclosure.** This work was conducted at Microsoft Research Asia. It does not evaluate proprietary products or services of Microsoft, and the research conclusions were not influenced by commercial interests.

## 2. Related Work

**Reasoning-Based Layout Planning.** Grounding spatial reasoning into metric layouts remains hindered by fragmented workflows. Multi-stage pipelines (Bucher & Armeni, 2025; Yang et al., 2025c; Ran et al., 2025) decouple logic from generation, leading to rigid autoregressive ordering (Bucher & Armeni, 2025), semantic-geometric mismatch (Yang et al., 2025c), or lost stacking relations (Ran et al., 2025), while reinforcement learning (RL) baselines (Pan & Liu, 2025) suffer from exploration inefficiency. Direct generation exposes abstract-to-metric gaps (Feng et al., 2023) or initialization sensitivity (Sun et al., 2025a; Wong et al., 2025), and agentic systems (Yang et al., 2025b; 2024c; Çelen et al., 2024) often fragment semantics across multi-step refinement. Furthermore, other methods exhibit domain rigidity (Hu et al., 2025; Sun et al., 2025c) or accumulate pixel-level inconsistencies into warped shells (Sun et al., 2025b). SceneReVis (Anonymous, 2026) uses a multi-turn RL formulation to improve generation quality and enable broader applications. In

contrast, our Zone-Graph Paradigm keeps zone-level structure explicit, linking intent, topology, and metric constraints to ensure global coherence under irregular boundaries.

**Data-Driven 3D Indoor Scene Generation.** Existing generators struggle to align optimization objectives with scene semantics. Autoregressive frameworks (Paschalidou et al., 2021) accumulate irreversible placement errors due to missing global planning, while specialized baselines (Tang et al., 2024; Yang et al., 2024a) relying on implicit priors fail strict non-convex boundaries or prioritize local physics over global function. Optimization-driven methods also exhibit gaps: LLplace (Yang et al., 2024b) lacks mechanisms for global functional structure, while RL-based methods like MetaSpatial (Pan & Liu, 2025) and ReSpace (Bucher & Armeni, 2025) succumb to reward hacking driven by discriminator inaccuracies or rigid manual rules. Furthermore, pipelines adopting Direct Preference Optimization (DPO) (Rafailov et al., 2023) such as OptiScene (Yang et al., 2025c) and DirectLayout (Ran et al., 2025) tend to overfit canonical patterns, restricting adaptive diversity. In contrast, Zone-Maestro interleaves reasoning internalization with Zone-Aware GRPO to jointly enhance semantic coherence and geometric consistency.

**3D Indoor Scene Datasets & Evaluation Protocols.** Data-driven synthesis relies on synthetic repositories, yet 3D-FRONT (Fu et al., 2021) and Structured3D (Zheng et al., 2019) are constrained by sparse arrangements, restricted typologies, or limited scale, while massive aggregations (Jia et al., 2024; Zhong et al., 2025) integrate diverse 3D scenes and assets (Zheng et al., 2019; Xiang et al., 2020; Dai et al., 2017; Chang et al., 2017; Baruch et al., 2021; Wald et al., 2019; Deitke et al., 2023) but suffer from format heterogeneity and physical defects. This deficit extends to evaluation protocols which remain confined to simplified settings: InstructScene (Lin & Mu, 2024) relies on canonical rectangular layouts, MesaTask (Hao et al., 2025) targets local tabletop rearrangement, and M3DLayout (Zhang et al., 2025) assesses semantic consistency without probing dense inter-object coupling. Furthermore, methods like PhyScene (Yang et al., 2024a) and SceneEval (Tam et al., 2026) evaluate semantic reasoning and geometric constraints as separable factors. This motivates SCALE, a benchmark designed to evaluate high-density spatial orchestration under complex boundary profiles and stylized semantic constraints across diverse room typologies.

# 3. Methodology

This section presents ZoneMaestro, a unified framework that internalizes the Zone-Graph Paradigm shown in Figure 2. We first model **Zone-Graph Orchestration** in Section 3.1 and describe Zone-Scene-10K in Section 3.2. We then introduce Alternating Spatial Alignment in Section 3.3, which interleaves Zone Reasoning Internalization with geometric

denoising via Zone-Aware GRPO (Z-GRPO).

## 3.1. Problem Formulation

Central to our approach, we formalize **Zone-Graph Orchestration** as a conditional generation problem. Given a natural language instruction $\mathcal{X}$, the model outputs a physically valid 3D spatial configuration $\mathcal{S}$. Unlike standard layout tasks that populate a fixed room, our setting requires the model to infer the architectural envelope from the internal zone structure and spatial relations.

We represent the target scene $\mathcal{S}$ as a compositional tuple $\mathcal{S} = (\mathcal{D}, \mathcal{G}, \mathcal{T}, \mathcal{A})$. The *Zone Inventory* $\mathcal{D}$ defines functional zones and their asset catalogs. The *Intra-Zone Spatial Graph* $\mathcal{G}$ captures local subgraphs where nodes are assets and edges encode spatial constraints. For example, Sofa $\xrightarrow{\text{anchor}}$ Coffee Table describes an anchor relation. The *Global Topology* $\mathcal{T}$ specifies inter zone adjacency. For example, Dining Zone $\xrightarrow{\text{north of}}$ Kitchen Zone encodes a relative placement. The *Architecture* $\mathcal{A}$ denotes the derived boundary polygon that encapsulates the assembled topology.

To model the Zone-Graph Paradigm, we factorize the joint probability distribution to reflect this *compositional inference* flow. Local designs determine global topology, which in turn dictates the architecture.

$$
\begin{aligned}
P(\mathcal{S}, \mathcal{R}|\mathcal{X}) = &P(\mathcal{A}|\mathcal{T}, \mathcal{G}) \cdot P(\mathcal{T}|\mathcal{G}, \mathcal{R}_{\text{topo}}) \\
&\cdot P(\mathcal{G}|\mathcal{D}, \mathcal{R}_{\text{spatial}}) \cdot P(\mathcal{D}|\mathcal{X}, \mathcal{R}_{\text{design}})
\end{aligned}
\tag{1}
$$

The generation follows a *Design Monologue* $\mathcal{R} = \{\mathcal{R}_{\text{design}}, \mathcal{R}_{\text{spatial}}, \mathcal{R}_{\text{topo}}\}$ that enforces a causal order. The model first determines zone contents $\mathcal{D}$. It then arranges intra-zone assets $\mathcal{G}$, assembles zones into a cohesive whole $\mathcal{T}$, and finally derives the enclosing boundary $\mathcal{A}$. This bottom-up factorization ensures that layouts follow semantic requirements rather than arbitrary placement. A complete input-output example is provided in Section D.4.

## 3.2. Dataset Construction: Zone-Scene-10K

To support the Zone-Graph Paradigm, we construct Zone-Scene-10K, a large-scale dataset built upon Intern-Scenes (Zhong et al., 2025) and enriched with explicit synthesized Zone-Graph annotations and Reasoning Monologues, as illustrated in Figure 2. By explicitly grounding latent spatial logic into the data, we provide the necessary supervision for determining functional zones, topological dependencies, and architectural boundaries. The end-to-end construction procedure is summarized in Algorithm 1 and Section B.2.

**Zone-Graph Annotation Pipeline.** Since raw layouts lack functional grouping, we build a Visual-Semantic Decomposition Pipeline that recovers zones and their constraints. Given multi-view renderings, GPT-4o (Hurst et al., 2024) clusters assets into candidate zones and we refine the clusters with split/merge heuristics. For each zone, we

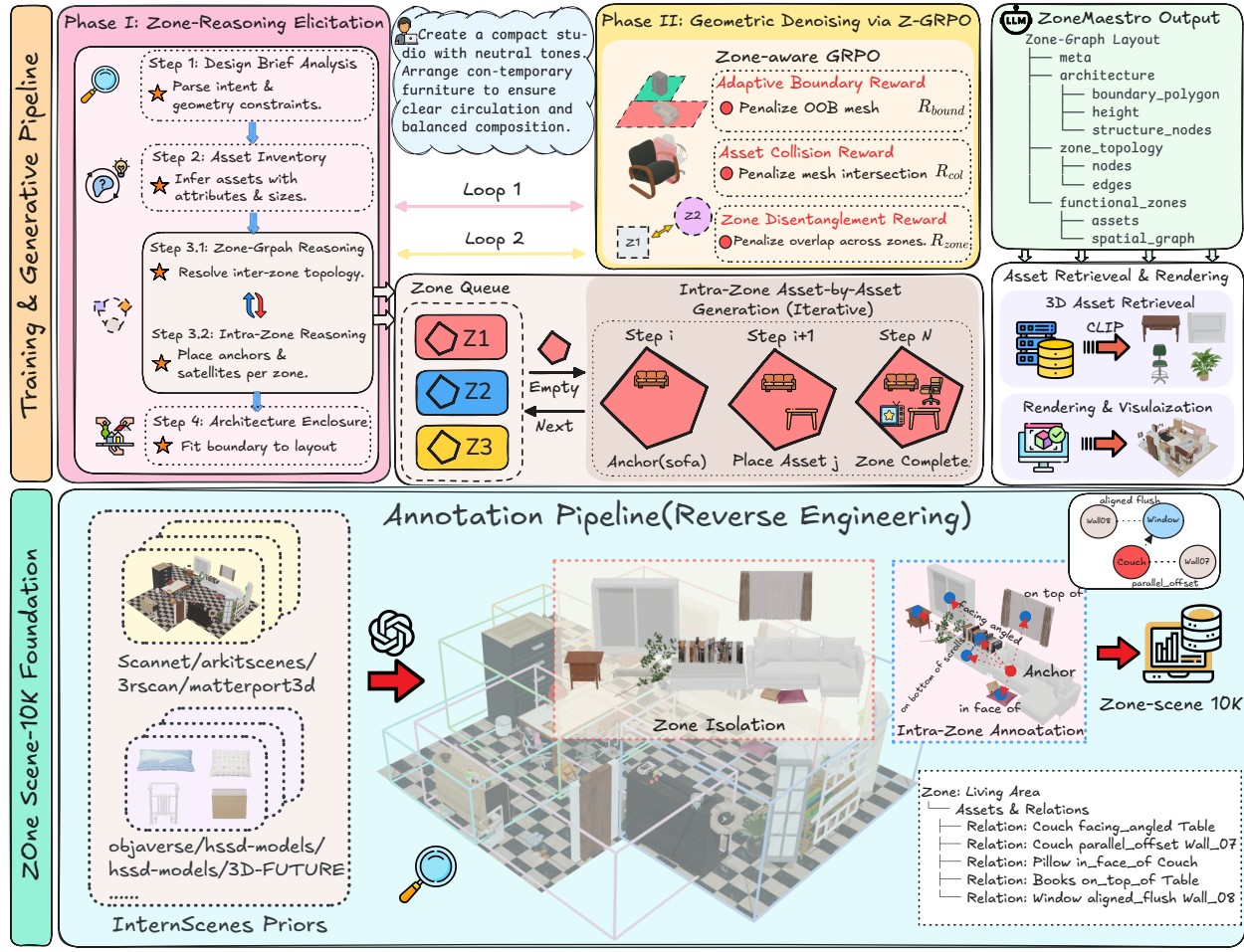

*Figure 2.* **The Zone-Graph Paradigm.** Our framework decomposes complex scene generation into four stages: (a) Zone Inventory, (b) Intra-Zone Spatial Graph, (c) Global Topology, and (d) Architecture Derivation.

render a masked view to extract the intra-zone graph $\mathcal{G}$, and we annotate the global topology $\mathcal{T}$ from zone adjacency and flow. We then generate a Zone-Graph Derivation $\mathcal{R}$ that narrates the same design → layout → architecture order used by our factorization.

**Synthesis of Multi-Granular Design Intents.** To reflect real user variability, we synthesize instructions at three granularities: *Coarse* descriptions capture overall atmosphere and intent, *Medium* prompts specify category-level furniture lists and composition, and *Fine* prompts state explicit geometric constraints. We further diversify styles and use GPT-4o to generate instructions from rendered views. Full granularity and style definitions are provided in Section B.2.

**Reverse-Engineering the Zone-Graph Derivation.** We synthesize the Zone-Graph Derivation $\mathcal{R}$ by reverse-engineering the ground-truth layout so the trace matches the granularity of $\mathcal{X}$: coarse prompts expand missing inventory and intent, while fine prompts emphasize constraint checking. This produces supervision that ties language, graphs, and geometry at a consistent level of detail.

### 3.3. Alternating Spatial Alignment

We propose Alternating Spatial Alignment, a cyclic optimization that alternates between (i) Reasoning Internalization, which teaches explicit Zone-Graph logic, and (ii) Geometric Denoising, which fixes physical violations via Zone-Aware GRPO. Reciprocal distillation mitigates reward-hacking drift and preserves lived-in realism; see Section A for theoretical motivation.

**Zone Reasoning Internalization.** Unlike flat generation that directly emits object coordinates, we train the model to output a structured reasoning trace that mirrors the design hierarchy. We use coarse-to-fine SFT: the model first predicts the zone inventory and global topology, then instantiates intra-zone graphs into concrete coordinates and orientations. This encourages global planning before local placement.

**Geometric Denoising via Zone-Aware GRPO.** We frame geometric refinement as constraint satisfaction solved by Group Relative Policy Optimization. Rather than exploring arbitrary behaviors, Z-GRPO acts as a denoising step that tightens physical compliance with a staged reward.

We first enforce boundary adherence to ensure containment of assets within the architectural envelope. To handle non-convex room shapes, we voxelize the room envelope induced by the decomposed maximal rectangles and penalize occupied asset voxels that protrude outside it:

$$R_{\text{bound}}(Y) = -\lambda_1 \sum_{o_j \in Y} \text{Vol}\big(\mathcal{V}(o_j) \setminus \Omega_{\mathcal{A}}\big),$$

$$\Omega_{\mathcal{A}} = \text{Voxel}\Big(\big(\bigcup_{m_k \in \mathcal{M}} m_k\big) \times [0, H]\Big) \tag{2}$$

where $\mathcal{V}(o_j)$ denotes the occupied voxel set of asset mesh $o_j$, and $\Omega_{\mathcal{A}}$ denotes the voxelized architectural interior.

Second, we maintain separation of functional groups at the intermediate level. We apply a zone disentanglement reward that penalizes the intersection of convex hulls belonging to different zones:

$$R_{\text{zone}}(Y) = -\lambda_2 \Big( \sum_{z_a \neq z_b} \text{IoU}(\mathcal{H}_{z_a}, \mathcal{H}_{z_b}) + \sum_z \text{Area}(\mathcal{H}_z \setminus \mathcal{P}) \Big) \tag{3}$$

Third, we resolve detailed physical intersections at the fine level. The asset collision reward imposes a volumetric penalty on intersections between voxelized asset meshes:

$$R_{\text{col}}(Y) = -\lambda_3 \sum_{i \neq j} \text{Vol}(\mathcal{V}(o_i) \cap \mathcal{V}(o_j)) \tag{4}$$

Naturally valid overlaps such as chairs under tables or cushions on sofas produce zero penalty when their occupied voxels do not intersect.

Finally, we drive this targeted relaxation via the composite reward $R = R_{\text{fmt}} + R_{\text{bound}} + R_{\text{zone}} + R_{\text{col}}$:

$$\mathcal{L}_{\text{GRPO}} = \mathbb{E}_{\mathcal{X}} \Big[ \frac{1}{G} \sum_i \min\big(r_i \hat{A}_i, \bar{r}_i \hat{A}_i\big) - \beta D_{\text{KL}} \Big] \tag{5}$$

where $\bar{r}_i = \text{clip}(r_i, 1-\epsilon, 1+\epsilon)$.

**Zone-Graph Evolution.** We integrate these two phases into a unified training loop. Each cycle begins with supervised internalization to establish a semantic prior. We then apply geometric alignment to refine physical coordinates. The improved layouts from the reinforcement learning phase are filtered and fed back as training signals for the next round of supervised internalization. This reciprocal distillation improves geometric precision while retaining semantic diversity from expert demonstrations. A theoretical motivation is provided in Section A. This loop stabilizes alignment so that optimization does not drift away from the user instruction.

# 4. Benchmarking Intricate Orchestration

We now address the critical lack of rigorous evaluation protocols for this domain. Existing benchmarks are saturated with canonical, rectangular layouts that fail to probe a model's ability to orchestrate architectural intricacy (Lin & Mu, 2024; Hao et al., 2025; Zhang et al., 2025; Yang et al., 2024a; Tam et al., 2026). To bridge this gap, we formalize the task of **Intricate Spatial Orchestration** and introduce **SCALE**, a synthetic evaluation suite constructed via a *Visually-Grounded Genesis Pipeline*. The full pipeline figure is in Figure 4 and Section C.

## 4.1. Task Definition

To ground subsequent evaluation, we define **Intricate Spatial Orchestration** as generating a full-room layout under two coupled difficulties: (i) geometric irregularity from non-convex boundaries, and (ii) semantic entanglement from high-density constraints with inter-object dependencies. Success requires satisfying both physical feasibility and global functional coherence, rather than optimizing object placement in a canonical convex shell.

## 4.2. The SCALE Benchmark

Building on this task definition, we introduce **SCALE** for rigorous evaluation. A key challenge is the *grounding gap*: text-only prompt design can be linguistically plausible yet geometrically infeasible. SCALE addresses this by constructing instructions from visually verified layouts, using generative vision systems as a physical plausibility filter. The full pipeline figure is in Figure 4.

Concretely, the construction pipeline proceeds through three stages. (1) **Generation**: we synthesize 22,050 floor plans with the LongCat Image Generator (Ma et al., 2025), spanning 9 boundary types including Rectangular, L-shaped, T-shaped, U-shaped, H-shaped, Trapezoidal, a diagonal wall cut, a protruding nook, and other irregular variants as shown in Figure 1. (2) **Inversion**: we reverse-engineer intent from each image via GPT-4o-mini (Hurst et al., 2024), producing an initial pool of 22,050 instructions. (3) **Curation**: we filter by format, deduplicate with CLIP to improve diversity, and stratify sampling, yielding 824 benchmark instances. Construction details appear in Section C.

# 5. Experiments

## 5.1. Experimental Setup

**Datasets.** We use **Zone-Scene-10K** for training and test, where all scenes are sourced from InternScenes(Zhong et al., 2025). User design instructions and Zone-Graph chains are generated following Section 3.2. Each SFT example contains a user design instruction and a reasoning chain with a Zone-Graph layout. We use $N = 8,500$ for SFT training, $N = 500$ for validation, and $N = 1,000$ for test, with training and validation plus test sampled from disjoint InternScenes sources with uniform room type coverage. For Z-GRPO, we sample $N = 5,120$ examples from the SFT training set, with $N = 2,560$ per cycle. We also evaluate

*Table 1.* Quantitative comparison on the Zone-Scene-10K Test Set and SCALE Benchmark. We evaluate physical validity, semantic quality, and system efficiency. **Bold** indicates the best performance. GPT-4o-mini scores are reported on a 1–10 scale where higher is better. Human reports overall preference rankings from our preference study where 1 is best and lower is better.

| Method | Physical Validity | | | Quality Assessment | | | | | | | Efficiency | |
|---|---|---|---|---|---|---|---|---|---|---|---|---|
| | OOB↓ | Col.↓ | Cnt | Aes.↑ | Real.↑ | Str.↑ | Geo.↑ | Sem.↑ | Func.↑ | Human↓ | Succ.↑ | Calls↓ |
| *Zone-Scene-10K Test Set* | | | | | | | | | | | | |
| DiffuScene (Tang et al., 2024) | 0.33 | 0.68 | 9.88 | 6.94 | 3.21 | 2.55 | 6.19 | 3.74 | 4.89 | 4.88 | 100% | – |
| ReSpace (Bucher & Armeni, 2025) | 0.15 | 0.10 | 4.36 | 7.23 | 3.71 | 4.74 | 7.80 | 7.63 | 6.99 | 4.36 | 99.9% | 5.3 |
| LayoutGPT (Feng et al., 2023) | 1.00 | 0.18 | 7.44 | 5.24 | 2.48 | 3.38 | 6.34 | 6.05 | 5.39 | 6.15 | 100% | 1 |
| LayoutVLM (Sun et al., 2025a) | 0.19 | 0.05 | 17.40 | 6.89 | 3.85 | 3.60 | 6.78 | 6.31 | 6.02 | 3.12 | 91.5% | 3 |
| Holodeck (Yang et al., 2024c) | 0.49 | 0.06 | 23.78 | 6.21 | 5.09 | 5.28 | 6.48 | 6.11 | 5.46 | 2.83 | 85.8% | 6.0 |
| i-Design (Çelen et al., 2024) | 2.80 | 2.34 | 15.30 | 6.00 | 3.21 | 3.50 | 5.62 | 5.04 | 5.11 | 5.09 | 75.6% | 9.3 |
| **ZoneMaestro** | **0.05** | **0.03** | 15.51 | **8.21** | **4.89** | **5.61** | **8.33** | **8.04** | **8.14** | **1.57** | **99.5%** | **1** |
| *SCALE Benchmark* | | | | | | | | | | | | |
| DiffuScene (Tang et al., 2024) | 0.22 | 0.74 | 10.20 | 7.45 | 3.36 | 2.92 | 6.37 | 4.43 | 5.27 | 5.47 | 100% | – |
| ReSpace (Bucher & Armeni, 2025) | 0.39 | 0.66 | 9.48 | 7.34 | 4.22 | 4.68 | 6.79 | 5.78 | 6.06 | 4.62 | 100% | 10.7 |
| LayoutGPT (Feng et al., 2023) | 6.17 | 0.72 | 14.47 | 5.51 | 2.71 | 3.26 | 4.85 | 4.39 | 4.04 | 5.51 | 100% | 1 |
| LayoutVLM (Sun et al., 2025a) | 0.27 | 0.06 | 20.16 | 6.28 | 3.58 | 3.64 | 5.95 | 4.84 | 5.07 | 3.74 | 94.4% | 3 |
| Holodeck (Yang et al., 2024c) | 0.74 | 0.05 | 21.90 | 4.64 | 3.59 | 3.95 | 5.14 | 3.95 | 3.77 | 2.96 | 72.1% | 8.4 |
| i-Design (Çelen et al., 2024) | 1.93 | 4.00 | 15.40 | 5.18 | 2.71 | 2.82 | 5.35 | 5.05 | 4.76 | 4.18 | 68.4% | 9.5 |
| **ZoneMaestro** | **0.09** | **0.04** | 23.35 | **7.88** | **4.95** | **5.19** | **8.22** | **7.82** | **7.69** | **1.52** | **98.7%** | **1** |

on the **SCALE Benchmark** with $N = 824$ instances, and both test protocols provide only user design instructions.

**Implementation Details.** We use Qwen3-8B (Yang et al., 2025a) and train on 8 NVIDIA A100 GPUs. Training follows two cycles of Alternating Spatial Alignment. For internalization, we run SFT for 2 epochs per cycle with global batch size 8 and 8 gradient accumulation steps. For geometric alignment, we run Z-GRPO with batch size 32 for 40 optimization steps. We use AdamW with learning rates $1e^{-5}$ for SFT and $5e^{-6}$ for Z-GRPO, with KL coefficient $\beta = 0.04$.

**Baselines.** We compare ZoneMaestro against: (1) **Data-Driven Methods** including DiffuScene (Tang et al., 2024) and ReSpace (Bucher & Armeni, 2025), using their official pre-trained checkpoints; and (2) **Agentic Frameworks** including LayoutGPT (Feng et al., 2023), LayoutVLM (Sun et al., 2025a), Holodeck (Yang et al., 2024c), and i-Design (Çelen et al., 2024), powered by GPT-4o (Hurst et al., 2024). We exclude OptiScene (Yang et al., 2025c) and DirectLayout (Ran et al., 2025) due to unavailable code and datasets, MetaSpatial (Pan & Liu, 2025) for requiring auxiliary scene information, and SceneWeaver (Yang et al., 2025b) given its substantial inference-time cost reported, making large-scale evaluation impractical.

**Evaluation Metrics.** We report five metric categories for systematic comparison. For **physical validity**, we measure Out-of-Bounds volume (OOB), collision volume (Col), and asset count (Cnt) for scene density. For **semantic quality**, a GPT-4o-mini judge rates layouts on a 1–10 scale across six dimensions: *Aesthetics* (Aes), *Realism* (Real), *Structure* (Str), *Geometry* (Geo), *Semantics* (Sem), and *Functional-*

*ity* (Func). To complement LLM-based scoring, we add **usability metrics** following SceneEval (Tam et al., 2026), computed from 2D occupancy grids at 0.05m resolution: navigability (NAV), walkable floor ratio (WFR), functional-side accessibility (ACC), minimum corridor width (MCW), collision-free reachability (REACH), and effective navigability $E\text{-}NAV = NAV \times \max(0, 1 - OOB)$. For **human evaluation**, we report both overall preference rankings (Human, lower is better) and a separate 30-evaluator instruction-following study over zone assignment, relative placement, and object combination, with Fleiss' $\kappa = 0.63$. For **efficiency**, we report generation success rate (Succ) and LLM inference calls per scene.

### 5.2. Main Results

We evaluate ZoneMaestro against leading baselines on both the held-out Test Set from Zone-Scene-10K and the SCALE Benchmark. Quantitative results are summarized in Table 1.

**Performance on Zone-Scene-10K.** The standard test set reveals a density-validity tension across all baselines. ReSpace (Bucher & Armeni, 2025) keeps violations low with OOB/collision 0.15/0.10, but generates only 4.36 assets per scene and scores 3.71 on realism, producing sparse and sterile rooms. DiffuScene (Tang et al., 2024) reaches 9.88 assets but suffers the worst collision volume at 0.68. Among agentic methods, i-Design (Çelen et al., 2024) collapses physically with OOB 2.80 and collision 2.34. LayoutGPT (Feng et al., 2023) also breaks boundaries with OOB 1.00 and has the lowest aesthetics at 5.24, while LayoutVLM (Sun et al., 2025a) shows weak zoning logic with structure 3.60. Holodeck (Yang et al., 2024c) reaches high

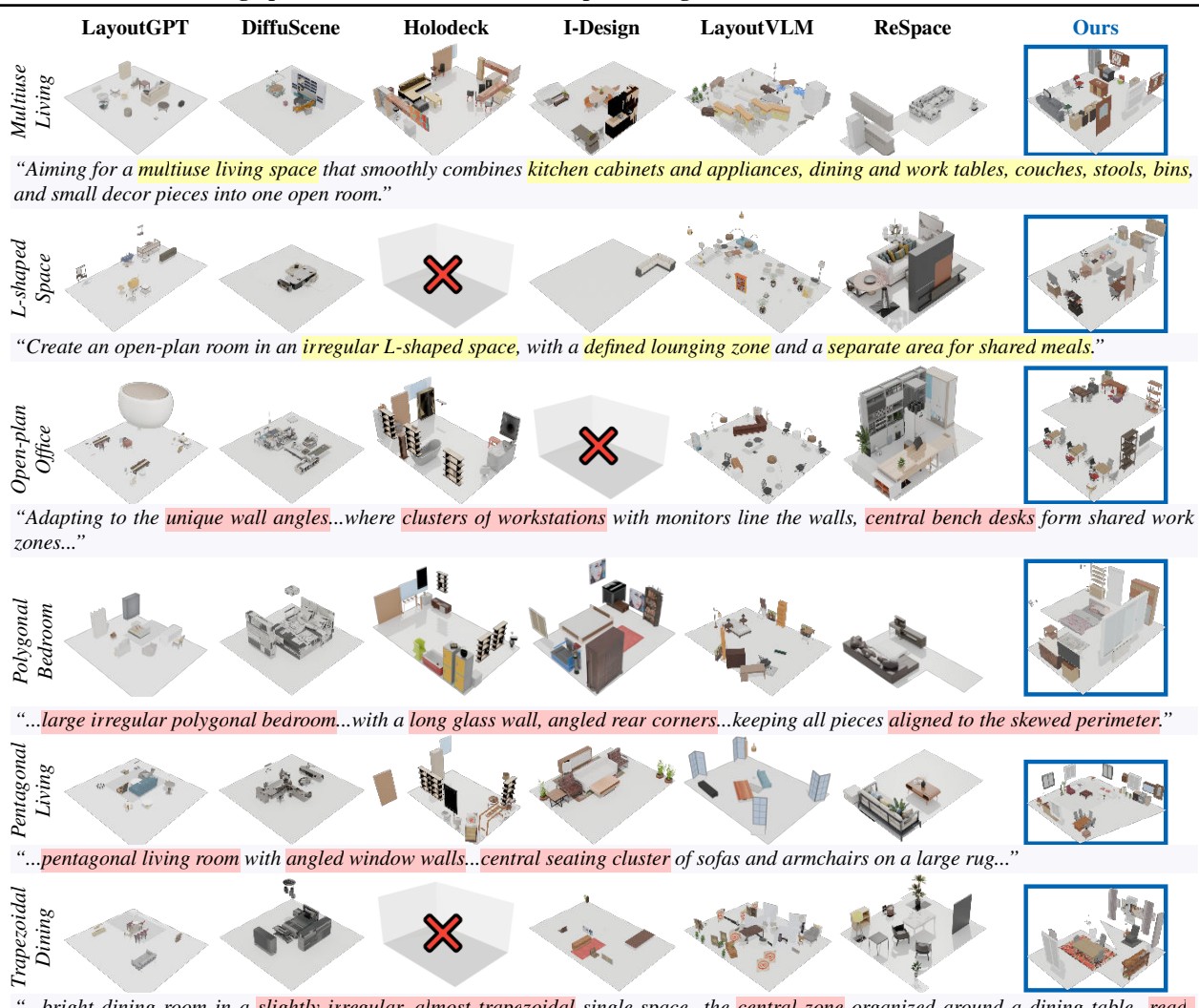

*Figure 3.* **Qualitative comparisons.** Rows 1–2: Zone-Scene-10K Test Set. Rows 3–5: SCALE Benchmark featuring complex non-convex geometries and dense asset arrangements. Highlighted instruction phrases often correspond to regions where layouts fail. Agentic frameworks that fail to manage high-density intersections are marked by ✗. ZoneMaestro follows fine-grained constraints while maintaining valid physical arrangements and alignment with irregular boundaries.

density at 23.78 assets but still sacrifices validity and tops out at 5.46 on functionality. ZoneMaestro is the only one that stays both dense and clean, generating 15.51 assets with OOB/collision 0.05/0.03, while ranking best on aesthetics, structure, and functionality.

*Table 2.* Scene usability metrics on SCALE. E-NAV combines navigability and boundary validity.

| Method | NAV | WFR | ACC | MCW | REACH | E-NAV |
|---|---|---|---|---|---|---|
| DiffuScene | 0.94 | 0.64 | 0.31 | 0.20 | 1.00 | 0.73 |
| ReSpace | 0.98 | 0.86 | **0.56** | **0.23** | 1.00 | 0.60 |
| LayoutGPT | 0.99 | **0.89** | 0.27 | 0.24 | 1.00 | 0.00 |
| LayoutVLM | 1.00 | 0.86 | 0.55 | 0.21 | 1.00 | 0.73 |
| Holodeck | **1.00** | 0.83 | 0.50 | 0.22 | 1.00 | 0.26 |
| i-Design | 0.96 | 0.76 | 0.31 | 0.20 | 1.00 | 0.00 |
| **ZoneMaestro** | 0.97 | 0.72 | 0.43 | 0.21 | 1.00 | **0.89** |

**Performance on the SCALE Benchmark.** SCALE is substantially harder due to non-convex shells and dense constraints. Methods stable on standard rooms degrade quickly: ReSpace (Bucher & Armeni, 2025) more than doubles its boundary errors, from 0.15 to 0.39; LayoutGPT (Feng et al., 2023) fails most dramatically with OOB surging from 1.00 to 6.17; i-Design (Çelen et al., 2024) loses control of intersections with collision volume reaching 4.00. ZoneMaestro remains stable, keeping OOB at 0.09 and collision at 0.04 while producing the highest asset density at 23.35. The quality gap is also structural rather than purely geometric: DiffuScene (Tang et al., 2024) drops to 2.92 on structure, and LayoutVLM (Sun et al., 2025a) falls to 3.58 on realism, revealing that complex boundaries amplify semantic fragmentation. Even Holodeck (Yang et al., 2024c), which keeps collision low, increases its boundary errors to 0.74

*Table 3.* Human instruction-following evaluation over 20 cases and 30 evaluators.

| Method | Zone | Rel. | Obj. | Avg. |
|---|---|---|---|---|
| DiffuScene | 35.8 | 41.3 | 53.2 | 43.4 |
| ReSpace | 49.5 | 55.2 | 64.8 | 56.5 |
| LayoutGPT | 22.4 | 37.5 | 72.1 | 44.0 |
| LayoutVLM | 54.6 | 60.8 | 78.4 | 64.6 |
| Holodeck | 61.2 | 65.4 | 82.6 | 69.7 |
| i-Design | 39.8 | 46.5 | 63.7 | 50.0 |
| **ZoneMaestro** | **88.5** | **84.2** | **92.3** | **88.3** |

and still fails to deliver usable dense arrangements.

**Usability-Oriented Evaluation.** To complement perceptual scores, Table 2 reports SceneEval-style metrics computed from 2D occupancy grids at 0.05m resolution. Several baselines obtain high local NAV because they leave large empty regions, but boundary violations sharply reduce effective navigability. ZoneMaestro achieves the highest E-NAV of 0.89, showing that its physical validity gains translate into usable layouts rather than only lower geometric error.

**Efficiency vs. Complexity Analysis.** Methods relying on test-time adaptation pay a large cost in repeated calls: ReSpace (Bucher & Armeni, 2025) and i-Design (Çelen et al., 2024) require 10.7 and 9.5 inference calls per scene on average. ZoneMaestro uses a single inference pass and achieves a 98.7% success rate because geometric reasoning is internalized during training.

### 5.3. User Study

We first conduct a preference study to quantify perceptual quality in irregular environments. 10 participants ranked seven methods by overall preference on 140 anonymized layouts, with 9 stratified SCALE instances covering all boundary types Section 4.2 and 5 randomly sampled scenarios from the Zone-Scene-10K test set per person. As reported in the Human column of Table 1, ZoneMaestro ranks first on SCALE with 1.52 and remains best on the Zone-Scene-10K test set with 1.57. Human preference is not identical to GPT trends: ReSpace (Bucher & Armeni, 2025) receives higher GPT realism and semantics than Holodeck (Yang et al., 2024c) on SCALE, yet Holodeck is ranked second by humans while ReSpace drops to 4.62.

**Instruction-Following Judgement.** We further expand the human evaluation to directly target instruction following. In Table 3, 30 evaluators judge 20 anonymized cases across zone assignment, relative placement, and object combination, with Fleiss' $\kappa = 0.63$. ZoneMaestro reaches 88.3% average correctness, exceeding Holodeck (Yang et al., 2024c) by 18.6 points and improving all three dimensions. This confirms that the gains are reflected in human judgments, not only to GPT-based scores.

### 5.4. Qualitative Analysis

We provide visual comparisons across two evaluation settings in Figure 3: the Zone-Scene-10K Test Set (Rows 1–2) evaluates generalization to diverse real world scenarios, while the SCALE Benchmark (Rows 3–6) targets intricate geometries. From both settings, we highlight three dimensions where ZoneMaestro improves over prior methods. See Section E.3 for additional qualitative cases.

**Density-Validity Breakthrough.** Current paradigms struggle to reconcile asset density with physical validity. In the Open-plan Office scenario, optimization-based agents like i-Design (Çelen et al., 2024) fail to converge on massive collision constraints and result in invalid states marked by the cross symbol. LayoutGPT (Feng et al., 2023) avoids conflicts by generating sparse and disconnected clusters. In the Multiuse Living scene the baselines miss the functional density required by the prompt. ZoneMaestro orchestrates over 20 assets in a single pass without collision. It forms distinct workstation clusters in the office and separates kitchen utilities from the dining zone in the living room. This confirms that the internalized Zone-Graph Paradigm effectively buffers the cognitive load of massive arrangements to maintain structural clarity.

**Geometric Intelligence in Non-Convex Spaces.** Irregular geometries expose the rigidity of heuristic planners. In Row 2 and Row 6, Holodeck (Yang et al., 2024c) fails to navigate the reentrant corners or narrowing widths. Similarly, ReSpace (Bucher & Armeni, 2025) and LayoutVLM (Sun et al., 2025a) struggle with the slanted perimeter in the Polygonal Bedroom in Row 4 by placing beds that intersect walls due to axis-aligned biases. ZoneMaestro exhibits precise geometric grounding by aligning large furniture strictly with adjacent wall normals. It utilizes irregular nooks for secondary functions and treats the boundary shape as a guiding constraint rather than an obstacle.

**Emergent Zone Topology.** ZoneMaestro demonstrates superior topological planning in eccentric spaces beyond obstacle avoidance. In Row 5, baselines like LayoutVLM (Sun et al., 2025a) and ReSpace (Bucher & Armeni, 2025) scatter objects randomly along the walls and rely on alignment heuristics that fail in pentagonal shapes. ZoneMaestro generates a coherent central seating cluster anchored by the rug independent of irregular wall angles. In Row 6, ZoneMaestro successfully distinguishes a reading nook from the primary dining area, while Holodeck (Yang et al., 2024c) fails on the tapered geometry. This behavior validates the Zone-Graph Paradigm, prioritizing functional connectivity over absolute coordinates to preserve human-centric circulation in atypical floor plans.

### 5.5. Ablation Studies

To validate the contribution of each component in our framework, we conduct ablation studies on the SCALE bench-

*Table 4.* Ablation Study on the SCALE Benchmark. We systematically analyze the contribution of each design choice including Zone-Graph Internalization and the Alternating Alignment strategy. GPT-4o-mini scores are reported on a 1 to 10 scale.

| Variant | Physical Validity | | | GPT-4o-mini Scores (1–10 scale) | | | | | | |
|---|---|---|---|---|---|---|---|---|---|---|
| | OOB ↓ | Col. ↓ | Count | Aes. ↑ | Real. ↑ | Struct. ↑ | Geo. ↑ | Sem. ↑ | Func. ↑ | Avg. ↑ |
| Base Reasoning SFT w/o Zone-Graph | 0.25 | 0.18 | 38.45 | 7.96 | 3.97 | 6.17 | 7.77 | 7.21 | 7.36 | 6.74 |
| Zone-Graph SFT Only | 0.21 | 0.13 | 30.96 | 7.97 | 3.89 | 6.23 | 8.19 | 7.26 | 7.50 | 6.84 |
| Z-GRPO Only w/o Alternating | 0.11 | 0.05 | 21.35 | 7.95 | 4.14 | 6.28 | 8.04 | 7.24 | 7.45 | 6.85 |
| Alternating Alignment One Cycle | 0.14 | 0.05 | 22.30 | 7.90 | 4.05 | 6.15 | 8.00 | 7.10 | 7.30 | 6.75 |
| Full Framework ZoneMaestro | 0.09 | 0.04 | 23.35 | 7.88 | 4.95 | 5.19 | 8.22 | 7.82 | 7.69 | 6.96 |

*Table 5.* Z-GRPO leave-one-out reward ablation on SCALE.

| Variant | OOB | Col. | Real. | Func. | Avg. |
|---|---|---|---|---|---|
| Full | **0.09** | **0.04** | **4.95** | **7.69** | **6.96** |
| w/o $R_{bound}$ | 0.15 | 0.05 | 4.78 | 7.59 | 6.84 |
| w/o $R_{zone}$ | 0.11 | 0.06 | 4.88 | 7.63 | 6.91 |
| w/o $R_{col}$ | 0.10 | 0.11 | 4.73 | 7.58 | 6.88 |

*Table 6.* Controlled ReSpace data comparison.

| Variant | OOB | Col. | Cnt | Aes. | Real. | Str. | Geo. | Sem. | Func. | Avg. |
|---|---|---|---|---|---|---|---|---|---|---|
| ReSpace orig. | 0.39 | 0.66 | 9.48 | 7.34 | 4.22 | 4.68 | 6.79 | 5.78 | 6.06 | 5.81 |
| ReSpace retrained | 0.24 | 0.25 | 9.07 | 7.48 | 4.52 | 4.90 | 7.15 | 6.12 | 6.35 | 6.09 |
| **ZoneMaestro** | **0.09** | **0.04** | **23.35** | **7.88** | **4.95** | **5.19** | **8.22** | **7.82** | **7.69** | **6.96** |

*Table 7.* Instruction-following stress test over 50 cases.

| Method | OOB | Col. | Cnt | Aes. | Real. | Str. | Geo. | Sem. | Func. | Avg. |
|---|---|---|---|---|---|---|---|---|---|---|
| DiffuScene | 0.29 | 0.61 | 8.94 | 6.62 | 2.85 | 2.68 | 5.82 | 3.82 | 4.58 | 4.39 |
| ReSpace | 0.52 | 0.73 | 8.31 | 6.55 | 3.75 | 4.45 | 6.28 | 5.22 | 5.45 | 5.28 |
| LayoutGPT | 8.05 | 0.64 | 13.82 | 4.88 | 2.32 | 2.95 | 4.40 | 3.82 | 3.52 | 3.65 |
| LayoutVLM | 0.20 | 0.10 | 21.87 | 5.58 | 3.15 | 3.38 | 5.48 | 4.28 | 4.50 | 4.40 |
| Holodeck | 0.58 | **0.05** | 19.21 | 4.08 | 3.12 | 3.65 | 4.65 | 3.45 | 3.28 | 3.71 |
| i-Design | 2.58 | 5.12 | 14.03 | 4.52 | 2.32 | 2.55 | 4.88 | 4.52 | 4.25 | 3.84 |
| **ZoneMaestro** | **0.11** | **0.05** | **23.91** | **7.00** | **4.27** | **6.11** | **7.62** | **7.38** | **7.03** | **6.57** |

2025) on Zone-Scene-10K using its original SFT+GRPO recipe; richer data improves its average score from 5.81 to 6.09, but it remains below ZoneMaestro and does not overcome the density ceiling. Table 7 stresses relative directions, corner-specific arrangements, and object combinations; ZoneMaestro achieves the best average score of 6.57 while keeping OOB/Col at 0.11/0.05.

mark. Results are detailed in Table 4 and Table 5.

**Internalized Reasoning as Structural Prior.** We assess Zone-Graph reasoning by training a variant *without Zone-Graph Derivation* that directly outputs layout JSON. In Table 4, it over-packs scenes with 38.45 assets, raising OOB/Col to 0.25/0.18 and reducing realism to 3.97. Zone-Graph SFT Only lowers OOB/Col to 0.21/0.13 by introducing zone-level allocation, showing that the structured trace suppresses over-packing and boundary drift before RL.

**Counteracting Reward Hacking.** We then study stability under reward optimization. Z-GRPO Only reduces collisions but produces cleaner, less expressive scenes, while one alternating cycle only partially restores density. Two full cycles recover 23.35 assets and the best **Realism (4.95)** while preserving geometric validity. The resulting realism–structure trade-off is analyzed in **Appendix D.5**.

**Reward Leave-One-Out.** Table 5 further isolates the three Z-GRPO rewards under the full alternating setup. Removing $R_{bound}$ increases OOB from 0.09 to 0.15, removing $R_{col}$ nearly triples collision from 0.04 to 0.11, and removing $R_{zone}$ weakens the semantic-function balance. The result supports the staged reward design: boundary, collision, and zone terms target complementary failure modes rather than redundant geometry penalties.

### 5.6. Additional Validation

**Data and Instruction Controls.** We close with two rebuttal controls. Table 6 retrains ReSpace (Bucher & Armeni,

## 6. Conclusion

We presented ZoneMaestro, a framework that internalizes Zone-Graph reasoning as a structural prior to preserve topological intent while adhering to non-convex geometric constraints. Our Alternating Alignment strategy effectively resolves the density-safety trade-off, bridging the gap between high-level planning and low-level physical execution. Furthermore, by releasing the SCALE Benchmark, we provide a rigorous testbed to advance 3D layout synthesis beyond canonical convex primitives.

## Impact Statement

This paper presents work whose goal is to advance the field of Machine Learning and 3D scene generation for embodied AI applications. We commit to releasing our code and the Zone-Scene-10K dataset upon acceptance to foster community collaboration and reproducibility. Beyond indoor scene synthesis, structured spatial orchestration may benefit urban planning, traffic scene generation for autonomous driving, and hierarchical GUI layout design, where dense constraints must be organized into coherent functional regions. These broader applications also require domain-specific validation because layout errors may affect accessibility, safety, or user experience. While enabling scalable environment synthesis,

this technology relies on training data that may reflect specific cultural or geographic architectural norms. There is a risk that the model propagates these biases by overrepresenting Western residential patterns while marginalizing diverse global living styles. We encourage practitioners to curate diverse datasets to mitigate such exclusion. Furthermore automated design tools carry implications for creative employment. We envision ZoneMaestro as an assistive system that enhances human productivity rather than replacing professional expertise. Users should exercise caution to prevent the generation of misleading virtual environments used for deceptive purposes.

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

This appendix is organized as follows. **Appendix A** provides theoretical motivation for the Alternating Alignment strategy. **Appendix B** details the Zone-Scene-10K dataset construction, including data sourcing, instruction synthesis, and statistics. **Appendix C** describes the SCALE benchmark construction pipeline. **Appendix D** presents implementation details such as the Zone-Graph annotation schema and training hyperparameters. **Appendix E** contains additional qualitative results and an extended case gallery. **Appendix F** provides the complete prompt collection used throughout the framework. **Appendix G** discusses limitations and future directions.

# A. Theoretical Motivation for Alternating Alignment

A natural question arises: why adopt an alternating alignment paradigm (Internalization $\rightarrow$ Alignment) rather than jointly optimizing semantic and geometric objectives from the start? We provide intuitive analysis grounded in gradient dynamics and generalization considerations.

## A.1. Gradient Conflict in Joint Optimization

**Observation 1 (Gradient Conflict).** *The gradients of semantic and geometric objectives tend to exhibit negative cosine similarity during early training, leading to destructive interference.*

Formally, let $g_{\text{sem}} = \nabla_\theta \mathcal{L}_{\text{sem}}$ and $g_{\text{geo}} = \nabla_\theta \mathcal{L}_{\text{geo}}$. We observe that:

$$\cos(g_{\text{sem}}, g_{\text{geo}}) = \frac{\langle g_{\text{sem}}, g_{\text{geo}} \rangle}{\|g_{\text{sem}}\|\|g_{\text{geo}}\|} < 0 \tag{6}$$

during the initial optimization phase. This occurs because the semantic objective encourages generating rich, diverse layouts (high entropy in asset placement), while the geometric objective penalizes any placement near boundaries or other objects, favoring sparse, conservative solutions. The resulting gradient $g_{\text{joint}} = g_{\text{sem}} + \lambda g_{\text{geo}}$ has reduced magnitude $\|g_{\text{joint}}\| < \|g_{\text{sem}}\|$, slowing convergence and often converging to suboptimal saddle points.

## A.2. Generalization Bounds via Curriculum Learning

**Intuition 2 (Alternating Alignment as Curriculum Learning).** *Decoupled training can achieve tighter generalization bounds by decomposing the hypothesis class.*

Consider the composite hypothesis class $\mathcal{H} = \mathcal{H}_{\text{sem}} \circ \mathcal{H}_{\text{geo}}$ where semantic reasoning precedes geometric grounding. By the PAC-Bayes framework, the generalization error of a composite learner satisfies:

$$\mathcal{E}_{\text{hier}} \leq \underbrace{\mathcal{E}(\mathcal{H}_{\text{sem}})}_{\text{Internalization}} + \underbrace{\mathcal{E}(\mathcal{H}_{\text{geo}}|\mathcal{H}_{\text{sem}})}_{\text{Alignment}} \tag{7}$$

where the conditional complexity $\mathcal{E}(\mathcal{H}_{\text{geo}}|\mathcal{H}_{\text{sem}})$ is significantly smaller than the joint complexity $\mathcal{E}(\mathcal{H})$ because the Alignment step operates on a *semantically anchored* representation rather than raw inputs. Intuitively, once the model has internalized Zone-Graph structure, the geometric refinement becomes a lower-dimensional optimization problem–adjusting coordinates within established functional zones rather than jointly discovering both semantics and geometry.

## A.3. Information-Theoretic Interpretation

From an information bottleneck perspective, Internalization compresses the instruction $\mathcal{X}$ into a condensed semantic representation $\mathcal{Z} = (\mathcal{D}, \mathcal{G}, \mathcal{T})$ that is maximally informative about the design intent while discarding geometric noise. The Alignment step then maps $\mathcal{Z} \rightarrow \mathcal{A}$ with geometric constraints, operating on a cleaner, lower-entropy input. This decoupled compression-then-refinement mirrors the rate-distortion optimal coding strategy:

$$I(\mathcal{X}; \mathcal{A}) \leq I(\mathcal{X}; \mathcal{Z}) + I(\mathcal{Z}; \mathcal{A}|\mathcal{X}) \tag{8}$$

Joint training conflates these information channels, forcing the model to simultaneously preserve semantic detail and satisfy geometric hard constraints—objectives that compete for representational capacity.

---

**Algorithm 1** Zone-Scene-10K Construction Pipeline

---

**Input:** raw layouts $\mathbb{L}_{raw}$, VLM $\mathcal{V}$
**Output:** annotated dataset $\mathcal{D}_{ZS}$
{RENDER: multi-view rendering; GROUP/IGRAPH/TOPO: VLM annotators; REFINE: split/merge heuristics}
{PROMPT/DERIVE: multi-granular instruction synthesis + reverse-engineered derivation}
$\mathcal{D}_{ZS} \leftarrow \emptyset$
**for** each layout $L \in \mathbb{L}_{raw}$ **do**
   {**S1: Visual-Semantic Decomposition**}
   $I \leftarrow \text{RENDER}(L)$
   $\mathcal{Z} \leftarrow \text{REFINE}(\text{GROUP}_{\mathcal{V}}(I))$
   {**S2: Zone-Graph Extraction**}
   **for** each zone $z \in \mathcal{Z}$ **do**
      $G_z \leftarrow \text{IGRAPH}_{\mathcal{V}}(I, z)$ {anchors + relations}
   **end for**
   $T \leftarrow \text{TOPO}_{\mathcal{V}}(\mathcal{Z})$
   $S \leftarrow (\mathcal{Z}, \{G_z\}_{z \in \mathcal{Z}}, T)$
   {**S3: Multi-Granular Intent + Derivation**}
   **for** $g \in \{\text{COARSE}, \text{MEDIUM}, \text{FINE}\}$ **do**
      $X \leftarrow \text{PROMPT}_{\mathcal{V}}(I, S, g)$
      $R \leftarrow \text{DERIVE}_{\mathcal{V}}(X, S)$
      $\mathcal{D}_{ZS} \leftarrow \mathcal{D}_{ZS} \cup \{(X, S, R)\}$
   **end for**
**end for**
**return** $\mathcal{D}_{ZS}$

---

## B. Zone-Scene-10K Dataset Details

This appendix provides comprehensive details of the Zone-Scene-10K dataset construction, including data sourcing and curation, instruction synthesis, and dataset statistics.

### B.1. Data Sourcing and Curation

We integrate raw scenes from InternScenes and 3D-FRONT to ensure authentic coverage. To target Intricate Spatial Orchestration, we apply a rigorous filtration strategy focusing on *Geometric Complexity* (prioritizing non-convex boundaries and multiple functional areas) and *Asset Density* (discarding sparse scenes), while ensuring balanced *Typology Coverage* across seven room categories to address the scarcity of service spaces.

### B.2. Instruction Synthesis Details

**Zone-Scene-10K Construction Pipeline.** Algorithm 1 summarizes the end-to-end dataset construction procedure.

To address the high variance in user information density, we stratify instructions into a Granularity $\times$ Style matrix:

- **Coarse (Atmospheric):** Describes only overall mood (e.g., "A cozy vibe") to train Design Inference for hallucinating necessary assets.

- **Medium (Categorical):** Identifies major furniture types to target Layout Composition.

- **Fine (Metric):** Imposes strict dimensional and spatial constraints to enforce Geometric Grounding.

We further inject Stylistic Diversity by randomizing syntactic openings (e.g., Imperative vs. Aspirational) and conditioning on specific aesthetic styles (e.g., Industrial $\rightarrow$ metal textures).

*Table 8.* Zone-Scene-10K dataset statistics by room type.

| Room Type | Train | Val | Test |
|---|---|---|---|
| Bedroom | 2,845 | 165 | 300 |
| Living Room | 2,512 | 146 | 280 |
| Kitchen | 1,256 | 73 | 140 |
| Dining Room | 1,024 | 60 | 115 |
| Study/Office | 892 | 52 | 100 |
| Bathroom | 512 | 30 | 45 |
| Other | 377 | 22 | 20 |
| **Total** | **9,418** | **548** | **1,000** |

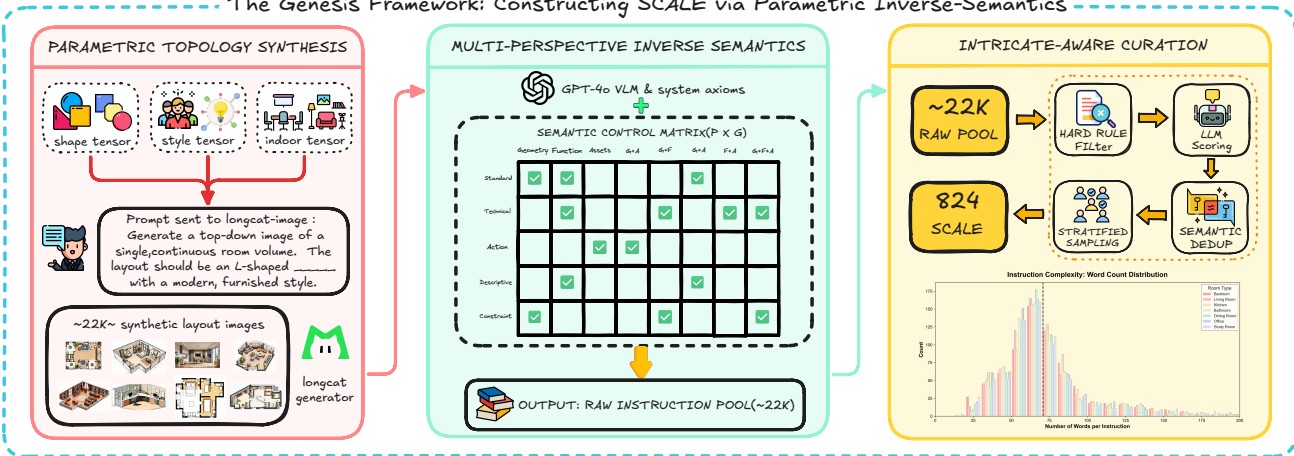

*Figure 4.* **The SCALE Benchmark Construction Pipeline.** We employ a parametric topology sampler to generate diverse non-convex boundaries, followed by a persona-driven inverse semantics engine to synthesize multi-granular instructions.

### B.3. Dataset Statistics

## C. SCALE Benchmark Construction Details

This appendix provides comprehensive details of the SCALE Benchmark construction, including all prompts used in image generation and instruction generation, as well as the complete data cleaning pipeline. We explicitly verify that the LongCat images and prompts used to construct SCALE are strictly disjoint from the InternScenes training set to prevent data leakage.

### C.1. Floor Plan Image Generation

**Image Generation Model** We use **LongCat-Image** (Ma et al., 2025), a 6B parameter text-to-image model from Meituan, optimized for bilingual (Chinese-English) text rendering, high photorealism output, and efficient inference (∼12GB VRAM with CPU offload).

**Generation Configuration    Room Shapes (9 types):** 1. Rectangular; 2. L-shaped; 3. T-shaped; 4. U-shaped; 5. H-shaped; 6. Trapezoidal; 7. Room with a diagonal wall cut; 8. Room with a protruding nook/alcove; 9. Other irregular shapes.

**Room Types (7 categories):** Bedroom, Living Room, Kitchen, Bathroom, Dining Room, Office, Study Room.

**Interior Styles (7 variations):**

1. Modern interior design, fully furnished with complete amenities

2. Comfortable lived-in atmosphere, well-organized layout

3. Functional layout with distinct activity zones and ample storage

4. Spacious arrangement with multiple furniture groupings

5. Contemporary style with detailed decor and accessories

6. High-efficiency layout maximizing floor space utility

7. Luxurious design with distinct separation of functions

**Repetitions:** 5 different seeds per combination. Total Images generated: $9 \times 7 \times 7 \times 10 \times 5 = 22,050$.

**Image Generation Prompts   Prefix (Shared across all images):**

"Generate a single high-quality room architectural 2D floor plan image, top-down vertical view, bird's eye view, orthographic projection, clean lines, flat shading, clearly defined walls, boundaries and furniture. The image depicts a single continuous open space defined by the outer perimeter walls only. There are absolutely no internal walls, no partitions, and no secondary rooms inside the boundary."

**Suffix Templates:** Each suffix template contains placeholders {shape} and {style} that are dynamically filled. We utilized distinct templates for each room type to ensure diversity.

*Bedroom Templates:*

1. "A {shape} bedroom layout. The sleeping zone is centered, with freestanding wardrobe units lining one wall. {style}."

2. "Plan of a {shape} single bedroom. A study desk is positioned near the window, sharing the open space with the bed. {style}."

3. "Top-down view of a {shape} bedroom. The room features a dressing area defined simply by a mirror and open clothing racks, not walls. {style}."

4. "A {shape} bedroom designed for two people. Twin beds are arranged symmetrically in the single open space. {style}."

5. "Layout of a {shape} bedroom where a lounge chair creates a reading nook in the corner of the room. {style}."

6. "A large {shape} master bedroom. A sofa sits at the foot of the bed, creating a sitting zone within the open floor plan. {style}."

7. "View of a {shape} bedroom with extensive storage cabinets arranged along the perimeter walls. {style}."

8. "A {shape} bedroom with an asymmetric furniture arrangement to fit the irregular wall geometry. {style}."

9. "A compact {shape} bedroom layout where the bed is tucked into a niche of the outer wall. {style}."

10. "A {shape} bedroom featuring a makeup station and dresser integrated into the main sleeping area. {style}."

*(Similar templates were used for other room types, focusing on their specific furniture and functional zones.)*

### C.2. Reverse Instruction Generation

**Overview**   We employ GPT-4o-mini to generate natural user instructions by analyzing the generated floor plan images. This "reverse engineering" approach creates diverse, realistic prompts.

### System Prompt for Instruction Generation

```
### IDENTITY & MISSION
You are an expert AI creating a high-quality dataset for 3D indoor scene generation.
Your goal is to **Simulate a User Instruction** based on a provided 2D floor plan image.

### CRITICAL TRUTH (The "Single Room" Axiom)
Before generating any text, analyze the image with these absolute rules:
```

1. **Single Volume:** The image depicts ONE continuous room (Bedroom, Kitchen, etc.).
2. **No Structural Partitions:** Internal lines are furniture (wardrobes, screens), NOT walls.
3. **No Sub-Rooms:** Never describe separate rooms like "en-suite" or "pantry". Everything is in the open plan.

### TASK PROTOCOL
You will be given:
1. **[TARGET PERSONA]:** A specific style of user (e.g., Casual, Technical).
2. **[TEMPLATE STARTER]:** The example phrase you can refer to under TARGET PERSONA.
3. **[CONTENT FOCUS]:** The specific aspect of the image to highlight (Geometric Shape, Functional Zones, or Asset Density).

### EXECUTION STEPS
1. **Adopt the Persona:** Look at the [TEMPLATE STARTER]. If casual, use simple words. If technical, use precise terms.
2. **Analyze the Focus:**
   - If Focus = **Geometry**: Describe the L-shape, T-shape, or irregular boundary.
   - If Focus = **Function**: Describe how furniture creates zones without walls.
   - If Focus = **Assets**: List specific furniture items and describe the density.
3. **Complete the Instruction:**
   - Start exactly with the [TEMPLATE STARTER].
   - Continue the sentence naturally to describe the image.
   - Ensure the final output is a coherent, single-sentence command or request.

### OUTPUT FORMAT
Return **ONLY** the final completed instruction string. Do not add quotation marks.

**Content Focus Categories**   We define 7 content focus modes based on three fundamental aspects: **G** (Geometry), **F** (Function), and **A** (Assets).

| Focus | Name | Description |
|-------|------|-------------|
| G | Geometry | Focus on room shape (L/T/H/Irregular) and boundaries. |
| F | Function | Focus on functional zones/activities without specific lists. |
| A | Assets | Focus on furniture lists, counts, and density. |
| G+F | Geo+Func | Combine geometric shape and functional zoning. |
| G+A | Geo+Assets | Combine room shape and asset details. |
| F+A | Func+Assets | Combine functional zoning and asset details. |
| G+F+A | Full Complex | Include all aspects. |

**User Persona Styles**   We employ 7 persona styles to ensure linguistic diversity:

- **Type 1: Standard Imperative** (e.g., "Design a layout for a...")

- **Type 2: Casual Conversational** (e.g., "I'm looking for a design for a...")

- **Type 3: Strictly Technical** (e.g., "Generate an orthographic projection of a...")

- **Type 4: Constraint-First** (e.g., "Without using any internal structural walls...")

- **Type 5: Action-Oriented** (e.g., "Arrange a complete furniture set within a...")

- **Type 6: Geometry-Conditional** (e.g., "Given the specific boundary shape...")

- **Type 7: Descriptive Vision** (e.g., "A detailed top-down architectural view of a...")

### C.3. Data Cleaning Pipeline

**Pipeline Overview**   The pipeline consists of 5 stages:

1. **Raw Generation:** 22,050 images → 22,050 raw instructions.

2. **Quality Evaluation:** GPT-4o-mini scoring with hard filters (Image Leak, Multi-Room, Template Violation, Length Checks).

3. **Semantic Deduplication:** Greedy deduplication using `text-embedding-3-large` with a cosine similarity threshold of 0.8.

4. **Data Augmentation:** Supplementing non-Geometry focus data from cache.

5. **Balanced Sampling:** Removing simple Rectangles and uniformly sampling irregular shapes to ensure difficulty.

**GPT Quality Evaluation System Prompt**

```
You are a data quality evaluator for a text-to-3D indoor layout generation benchmark.
Your task is to evaluate a user instruction (prompt) that was generated to describe
a 2D floor plan image.

### EVALUATION CRITERIA
**Hard Filters (REJECT if ANY is true):**
1. IMAGE_LEAK: Contains phrases like "This image shows", "In the image".
2. MULTI_ROOM: Mentions separate rooms like "en-suite", "pantry".
3. TEMPLATE_VIOLATION: Does not start with the provided template_starter.
4. TOO_SHORT: Less than 50 characters.
5. TOO_LONG: More than 800 characters.
6. ROOM_MISMATCH: Describes wrong room type.

**Quality Score (1-10):**
1-3: Poor; 4-5: Below Average; 6-7: Good; 8-9: Very Good; 10: Excellent.

### OUTPUT FORMAT (JSON only)
{
  "pass_hard_filter": true/false,
  "reject_reason": "NONE" or [REASON],
  "quality_score": 1-10,
  "brief_comment": "One sentence reason"
}
```

### C.4. Final Benchmark Statistics

**Total Instructions:** 824. **Composition:** 563 (68.3%) contain explicit Geometry constraints; 261 (31.7%) are control samples.

*Table 9.* Distribution by Content Focus in SCALE.

| Focus | Count |
|---|---|
| Geometry (G) | 235 |
| Geometry + Assets (G+A) | 149 |
| Geometry + Function (G+F) | 129 |
| Assets (A) | 104 |
| Function (F) | 82 |
| Function + Assets (F+A) | 75 |
| Full Complex (G+F+A) | 50 |

## D. Implementation Details

This appendix provides implementation details of the ZoneMaestro framework, including the Zone-Graph annotation schema and training hyperparameters.

*Table 10.* Distribution by Room Shape in SCALE before and after deduplication.

| Shape | Original | Deduped | Retention |
|---|---|---|---|
| H-shaped | 2450 | 384 | 15.7% |
| Rectangular | 2450 | 324 | 13.2% |
| L-shaped | 2450 | 151 | 6.2% |
| Other irregular shapes | 2443 | 137 | 5.6% |
| Room with a diagonal wall cut | 2450 | 92 | 3.8% |
| T-shaped | 2450 | 75 | 3.1% |
| Trapezoidal | 2450 | 67 | 2.7% |
| Room with a protruding nook/alcove | 2450 | 57 | 2.3% |
| U-shaped | 2450 | 37 | 1.5% |

## D.1. Zone-Graph Annotation Schema

We provide the detailed JSON schema used for Zone-Graph annotations in Zone-Scene-10K. Each scene is annotated with:

- **Zone Definitions:** A list of functional zones, each containing zone ID, zone type (e.g., "Sleeping", "Working", "Dining"), and a list of asset IDs belonging to that zone.

- **Intra-Zone Graph:** For each zone, a directed graph where nodes are assets and edges encode spatial relations (e.g., "left_of", "in_front_of", "on_top_of").

- **Inter-Zone Topology:** A graph connecting zone centroids with adjacency relations (e.g., "north_of", "adjacent_to").

- **Design Monologue:** A natural language narrative explaining the design rationale, generated via reverse-engineering from the ground truth layout.

## D.2. The Group-Relative Objective Mechanics

Unlike PPO which requires a separate Value Model (introducing training instability), GRPO estimates the baseline directly from the group mean of sampled outputs, making it highly efficient for our massive-asset generation task. For each instruction $\mathcal{X}$, we sample a group of $G$ outputs $\{Y_1, \ldots, Y_G\}$ from the reference policy $\pi_{\theta_{\text{old}}}$. The optimization objective is formulated to push the model towards layouts that are relatively better than the group average:

$$\mathcal{L}_{\text{GRPO}}(\theta) = \mathbb{E}_{\mathcal{X}, \epsilon} \left[ \frac{1}{G} \sum_{i=1}^{G} \min \left( r_i(\theta)\hat{A}_i, \text{clip}(r_i(\theta), 1-\epsilon, 1+\epsilon)\hat{A}_i \right) - \beta D_{\text{KL}} \right] \tag{9}$$

where $r_i(\theta) = \frac{\pi_\theta(Y_i|\mathcal{X})}{\pi_{\theta_{\text{old}}}(Y_i|\mathcal{X})}$ is the importance ratio. The advantage $\hat{A}_i$ is computed by normalizing the total reward $R_i$ within the group:

$$\hat{A}_i = \frac{R_i - \text{mean}(\{R_j\}_{j=1}^{G})}{\text{std}(\{R_j\}_{j=1}^{G})} \tag{10}$$

This mechanism allows the model to explore the geometric solution space around the semantic anchor provided by SFT, effectively "denoising" the layout distribution.

## D.3. Hyperparameter Settings

**Training Configuration.** We employ Qwen3-8B (Yang et al., 2025a) as our backbone foundation model, distributed across 8 NVIDIA A100 GPUs. Our training protocol follows the proposed Alternating Alignment strategy, executing two full cycles of supervised internalization followed by reinforcement alignment. For the Zone-Graph Internalization phases, we fine-tune the model for 2 epochs per cycle with a global batch size of 8 and 8 gradient accumulation steps. Subsequently, the Geometric Alignment phases employ Z-GRPO with a training batch size of 32 for 40 optimization steps. We use the AdamW optimizer with learning rates of $1e^{-5}$ for supervised phases and $5e^{-6}$ for Z-GRPO, setting the KL divergence coefficient $\beta = 0.04$.

*Table 11.* Hyperparameters for ZoneMaestro training.

| Parameter | Value |
|---|---|
| Backbone Model | Qwen3-8B (Yang et al., 2025a) |
| Hardware | $8\times$ NVIDIA A100 |
| Alternating Alignment Cycles | 2 |
| SFT Epochs per Cycle | 2 |
| Global Batch Size (SFT) | 8 |
| Grad Accumulation (SFT) | 8 |
| Z-GRPO Batch Size | 32 |
| Z-GRPO Optimization Steps | 40 |
| SFT Learning Rate | 1e-5 |
| Z-GRPO Learning Rate | 5e-6 |
| GRPO Group Size $G$ | 8 |
| KL Coefficient $\beta$ | 0.04 |
| $\lambda_1$ (Boundary) | 1.0 |
| $\lambda_2$ (Zone) | 0.5 |
| $\lambda_3$ (Collision) | 2.0 |
| Self-Distillation Rounds | 3 |

## D.4. Complete Input-Output Example

We present a complete input-output example from ZoneMaestro to illustrate the Zone-Graph reasoning process. The model receives a user instruction and generates a structured reasoning trace (`<think>`) followed by the final Zone-Graph layout (`<answer>`).

**User Instruction (Input).**

> *"Make a detailed room design for a living room that uses furniture placement to create distinct functional zones, including a central conversation and relaxation area, a clearly defined children's play and reading corner along one side, and smooth circulation paths that connect these activity zones within the single open space."*

**Zone-Graph Reasoning Trace (Output `<think>`).** The model first analyzes the instruction and inventories required assets with their dimensions and styles. It then plans four functional zones: (1) a central conversation area anchored by an L-shaped sectional couch with satellite seating and coffee table; (2) a children's play corner with a rocking horse and books; (3) a work/laptop zone with an office chair; and (4) a storage and decor wall with cabinets and windows. The reasoning trace explicitly considers zone topology, circulation paths, and boundary constraints before committing to coordinates.

**Zone-Graph Layout (Output `<answer>`).** The final output is a structured JSON containing:

- **Architecture**: Room boundary polygon (L-shaped, $4.78\text{m} \times 5.41\text{m}$), wall segments with normals, and window positions.

- **Zone Topology**: Four zones with inter-zone relations (`adjacent_open`, `anchored_against` walls).

- **Functional Zones**: Each zone contains assets with category, role (`zone_anchor` or `satellite`), position, rotation, size, and intra-zone spatial graph edges.

This example demonstrates how ZoneMaestro's Design Monologue explicitly reasons about zone contents $\mathcal{D}$, intra-zone graphs $\mathcal{G}$, global topology $\mathcal{T}$, and architecture $\mathcal{A}$ in a causal chain before producing the final layout.

## D.5. Analysis of Structure vs. Realism Trade-off

As noted in Section 5.5, ZoneMaestro achieves superior Realism (4.95) and Geometric Validity but scores slightly lower on the Structure metric compared to SFT baselines. To understand this discrepancy, we analyze the definition of the "Structure"

metric used in our GPT-4o evaluation prompt.

**The Definition of Structure.** The evaluation prompt for *Structural Logic* (Category 2) explicitly penalizes layouts that appear "scattered" or lack rigid grouping. As shown in the rubric below, the criterion favors high-level hierarchical zoning, which SFT baselines satisfy by generating sparse, grid-aligned arrangements.

---

GPT-4o Evaluation Rubric: Structural Logic

**Category 2: Structural Logic**
**3. Structural Orchestration (Critical)**

- **Focus:** Hierarchy & Grouping (Handling Massive Assets).

- **Criteria:** specifically for scenes with **massive assets (¿50 items)**, does the model organize them into logical functional groups/zones? Or are they scattered randomly/piled up?

- **Score (0-10):** 0 = Chaotic scattering; 10 = Clear, hierarchical zoning.

**4. Geometric Grounding (Critical)**

- **Criteria:** How well does the layout adapt to **irregular geometries**?

---

**Why ZoneMaestro Scores Lower.** SFT baselines, unconstrained by physical collision checks, often produce highly symmetric, grid-like patterns that visually maximize the "Hierarchical Zoning" score, despite lacking physical plausibility (Realism $\approx 3.9$).

In contrast, ZoneMaestro is optimized via RL to ensure **zero collisions** within complex non-convex boundaries. To accommodate high asset density (>50 items) without intersection, the model introduces **organic irregularities**—such as slightly rotating chairs to fit alcoves or creating asymmetric clusters. While these adjustments significantly enhance **Realism** and **Physical Validity**, they increase visual entropy, which the VLM judge partially misinterprets as a reduction in "clear hierarchical zoning." Thus, the lower Structure score reflects a shift from *artificial rigidity* to *organic, physically-grounded complexity*.

# E. Additional Qualitative Results and Extended Case Gallery

This appendix presents additional qualitative results and an extended case gallery demonstrating ZoneMaestro's capabilities across diverse room types and complexity levels. These examples further illustrate the advantages of Zone-Graph Orchestration in handling irregular geometries, maintaining zone coherence, and achieving lived-in realism.

## E.1. Additional Qualitative Comparisons

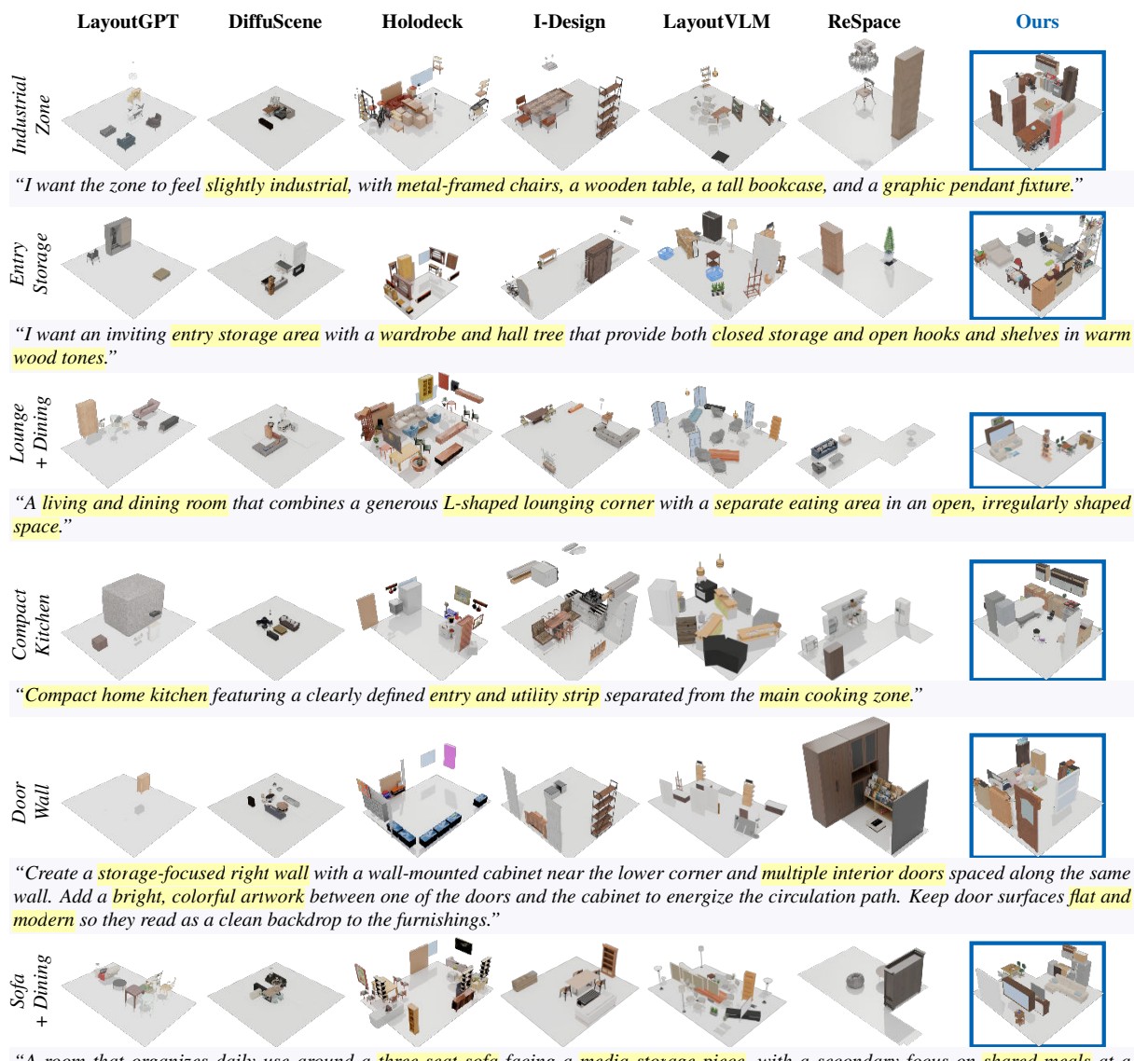

*"I want the zone to feel slightly industrial, with metal-framed chairs, a wooden table, a tall bookcase, and a graphic pendant fixture."*

*"I want an inviting entry storage area with a wardrobe and hall tree that provide both closed storage and open hooks and shelves in warm wood tones."*

*"A living and dining room that combines a generous L-shaped lounging corner with a separate eating area in an open, irregularly shaped space."*

*"Compact home kitchen featuring a clearly defined entry and utility strip separated from the main cooking zone."*

*"Create a storage-focused right wall with a wall-mounted cabinet near the lower corner and multiple interior doors spaced along the same wall. Add a bright, colorful artwork between one of the doors and the cabinet to energize the circulation path. Keep door surfaces flat and modern so they read as a clean backdrop to the furnishings."*

*"A room that organizes daily use around a three-seat sofa facing a media storage piece, with a secondary focus on shared meals at a centrally placed dining table further down."*

*Figure 5.* Additional qualitative comparisons with baselines on the Zone-Scene-10K Test Set (not shown in Figure 3). Full user instructions are provided without abbreviation.

| LayoutGPT | DiffuScene | Holodeck | I-Design | LayoutVLM | ReSpace | Ours |
|---|---|---|---|---|---|---|

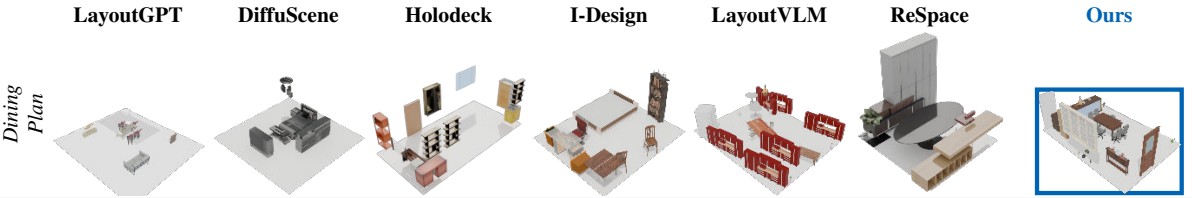

*"Visualize a technical top-down section of a single* open-plan dining room *where a* central rectangular dining table with six chairs *defines the primary eating zone, surrounded by* perimeter storage and display assets *including a long sideboard with décor, a tall crockery cabinet filled with dishes, an additional cabinet by the window, a low partition console with planters forming a subtle entry boundary, multiple potted plants in corners, and* large glazed doors *along one wall that function as the main light and access point."*

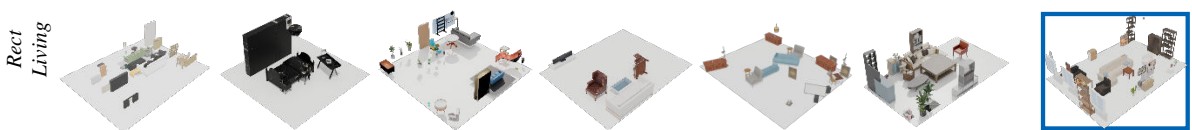

*"Configure the interior arrangement of a single,* spacious rectangular living room *with a clean open layout, placing a* large central seating zone *on a rug with two facing sofas, two armchairs, a pair of ottomans, and a nested coffee table set, oriented toward a TV console on one long wall, while using low bookshelves and storage units along the perimeter to subtly define reading and entertainment areas without partitions; add a* media wall *with a flat-screen TV, speaker, artwork, and tall plants near the corners, install a wall of windows with curtains on the opposite long side to bring in light and frame a small reading nook with an accent chair and side table, position tall bookcases and a dresser-style cabinet at one end to create a library/storage corner, line the remaining walls with additional shelving, plants, and decorative objects to keep asset density high but organized, and ensure all furniture is arranged to create* clear walking paths *around the perimeter and between zones while maintaining the room's* strong rectangular geometry*."*

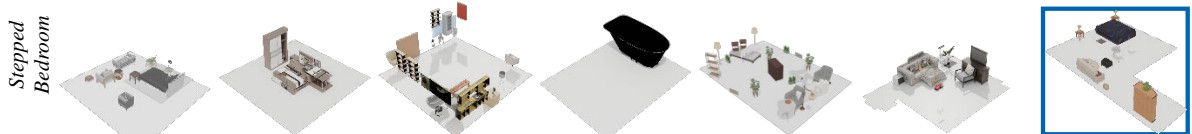

*"A flat-shaded architectural plan of a single* open-plan bedroom *with an* irregular, subtly stepped rectangular footprint*, where one long wall holds a* large bed centered on a rug *with matching nightstands and lamps, the opposite side features a* sofa and side table seating nook*, and the remaining floor area is partially filled with a dresser, small tables, accent chairs, plants, and layered rugs that visually partition the sleeping and lounging zones* without any interior walls*."*

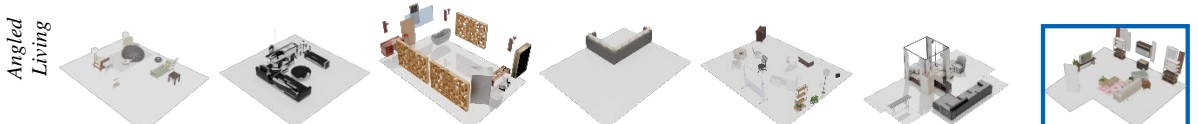

*"A complete interior design layout of a modern open-plan living room with an* irregular, slightly angled perimeter *and a* recessed entry area*, where an* L-shaped sectional sofa *and matching armchair create a cozy seating zone around a central coffee table and rug, a long TV console with a wall-mounted screen and nearby potted plant defines the* media zone by the entrance*, additional storage and display cabinets line the opposite wall under windows with curtains, and floor lamps, side tables, and other small decor pieces are arranged to balance the space and maintain* clear circulation paths *throughout the single continuous room."*

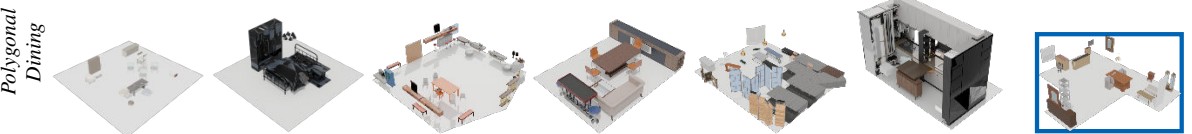

*"Produce a technical blueprint for a single* open-plan dining room *contained within one* irregular polygonal shell *with multiple angled exterior segments, clearly defining the* central circulation area *and stone-tiled floor pattern, a* primary dining zone *along one facet with a rectangular dining table and surrounding benches/chairs, a* secondary lounge/relaxation zone *with a low table and seating benches,* integrated linear storage and shelving units *along the perimeter walls, a* compact kitchen-style service counter with stools *and under-counter cabinets positioned off to one side to support dining functions, additional sideboards and console tables near the edges, and accurate placement, dimensions, and orientation for all furniture assets, doors, windows, and built-in elements so the entire multifunctional dining environment is represented as a* single continuous room *without internal partitions."*

*Figure 6.* Additional qualitative comparisons with baselines on the SCALE Benchmark (not shown in Figure 3). Full user instructions are provided without abbreviation.

### E.2. Full Instructions for Main Paper Figures

For reproducibility, we provide the complete, unabridged user instructions corresponding to the qualitative examples shown in the main paper. Figure 3 includes both the Zone-Scene-10K Test Set (Rows 1–2) and the SCALE Benchmark (Rows 3–5).

**Zone-Scene-10K Test Set (Rows 1–2)  Row 1 – Multiuse Living:**

*"Aiming for a multiuse living space that smoothly combines kitchen cabinets and appliances, dining and work tables, couches, stools, bins, and small decor pieces into one open room."*

**Row 2 – L-shaped Space:**

*"Create an open-plan room in an irregular L-shaped space, with a defined lounging zone and a separate area for shared meals."*

**SCALE Benchmark (Rows 3–5)  Row 3 – Open-plan Office:**

*"Can you help me arrange furniture in a single open-plan rectangular office like this, with four main workstation zones along the walls (each having a long wooden desk, rolling office chair, computer monitor, keyboard, mouse, desk lamp, plants, stationery pots, and small side drawers), an L-shaped corner workstation with overhead shelves, binders, books, lamps, and a small printer cabinet, plus wall-mounted pinboards and notes above each desk to define working areas, wide glass sliding doors on two adjacent walls for natural light, a rug in the center to mark a shared circulation/collaboration space, and enough open floor area in the middle for easy movement between all the workstations without adding any interior partitions?"*

**Row 4 – Polygonal Bedroom:**

*"Arrange a complete furniture set within a large irregular polygonal bedroom that widens toward the front with a long glass wall, angled rear corners, and a slightly tapered side, keeping all pieces aligned to the skewed outer perimeter."*

**Row 5 – Pentagonal Living:**

*"A high-quality 2D rendering showing a pentagonal living room with angled window walls and a mix of wood and stone flooring, where a central seating cluster of sofas, armchairs, and coffee table sits on a large rug, flanked by a long sideboard, numerous potted plants along the perimeter, and smaller accent tables that collectively fill and emphasize the unique faceted geometry of the space."*

## E.3. Extended Case Gallery

The following pages present extended qualitative results from our method.

**Zone-Scene-10K Test Set: Extended Cases.** We present additional cases from the Zone-Scene-10K test set below.

*I want a pendant lamp above the living area, centered roughly over the coffee table between the sofa and the middle of the room. This light should clearly mark the living zone and provide direct light to the seating cluster. It should hang in line with the main axis of the sofa and table.*

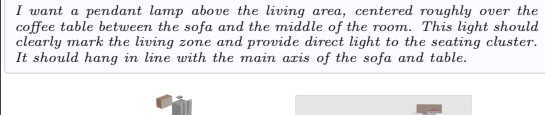

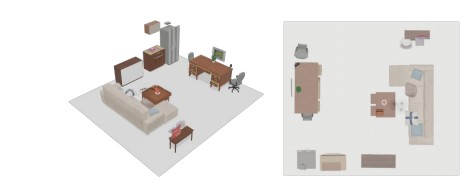

*Plan the overall layout so that larger pieces like sofa, dining table, tv stand, and sideboard are kept close to the walls or room centerlines, while smaller items such as side table, plant, and floor lamp fill in secondary positions. Maintain consistent spacing between furniture groups. Ensure each zone feels distinct yet visually connected.*

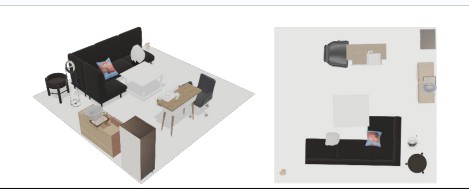

*Open rectangular gathering space featuring a symmetrical dining setup balanced by a linear media and seating layout.*

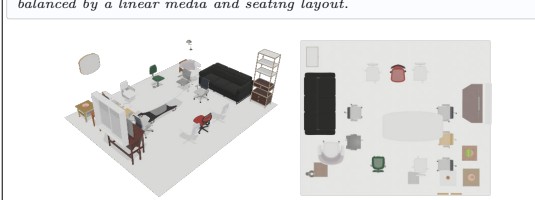

*Hoping to create a conversation-friendly living area where the sofa, armchair, and ottoman all loosely face the coffee table while still opening toward the dining zone. The sofa should hug the corner of the room, with a potted plant at one end and a side table at the other. A small storage cabinet can sit behind the armchair, helping to define the edge between living and circulation space.*

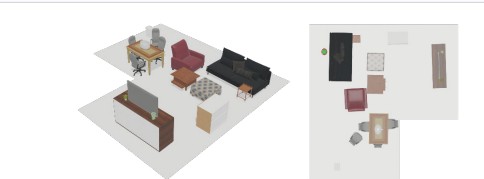

*Combined lounge and dining space featuring a low-profile media unit keeping the TV area visually light.*

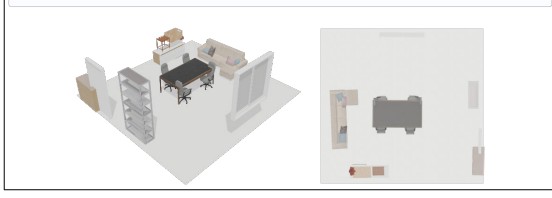

*Seeking a living-dining arrangement that runs lengthwise, with circulation maintained along the open side of the room.*

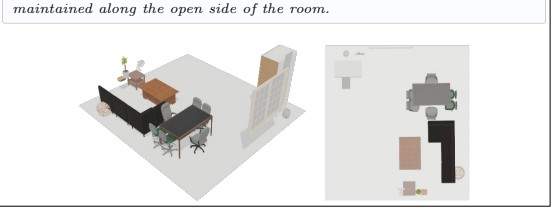

*Open-concept family living room emphasizing clear sightlines from the dining table across to the sofa and TV wall. Arrange all main furniture parallel to the room's long walls, avoiding diagonal placements that would disrupt the flow. Use the central open area around the pouf as flexible space for kids to play or guests to mingle.*

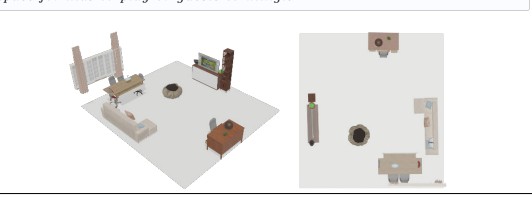

*Design the dining area so that when seated, diners on either side of the table face each other across the narrow width, making conversation easy. Orient the long dimension of the table along the room's length, reinforcing a formal, banquet-like arrangement. Use uniform high-back chairs to frame the table and add a sense of symmetry. Keep decorative elements low and centered to avoid visual clutter.*

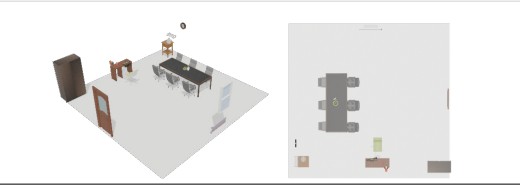

*A room that organizes a cozy sitting area around a TV and coffee table while centering a sizable dining table for six under its own overhead light.*

*Seeking a media wall that combines a tv_stand and sideboard with a plant nearby.*

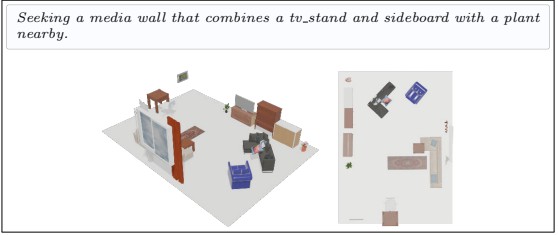

*A room that balances hard edges with soft details. The shelving, cabinets, and folders all present strong vertical and horizontal lines, while the curtain, hanging textiles, and draped shirt introduce fluid shapes along the walls. A rounded clock and a smooth pepper-shaped decor object break up the geometry further. This mix keeps the compact study from feeling too rigid.*

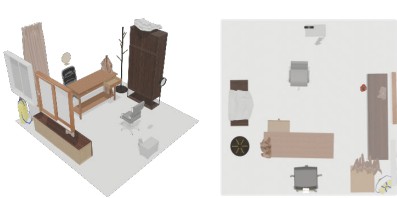

*Aiming for a tech and appliance cluster on and around the island that keeps the microwave, fruit, cans, and small accessories grouped on the counter and cabinet.*

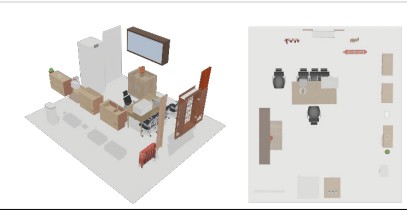

*I want a bedroom that supports both rest and light fitness, with the front wall devoted to a window, curtains, radiator, and a slim wall-mounted hanger. In front of that wall I'd like a squat rack, a treadmill positioned close to the right corner, and a digital scale close to the rack. The main bed should sit just behind this equipment, with a pillow at the side nearer the left. Opposite that, the stretcher bed should run along the back wall with a magazine holder and pillows.*

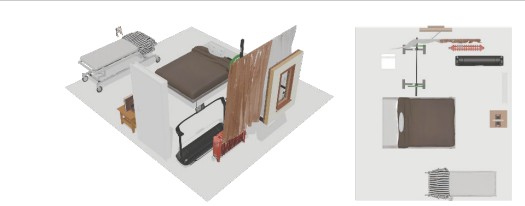

*A contemporary kitchen that balances a long service counter with integrated sink, wall cabinets, and small accessories like cups, bottles, and storage boxes in a warm wood-and-glass palette.*

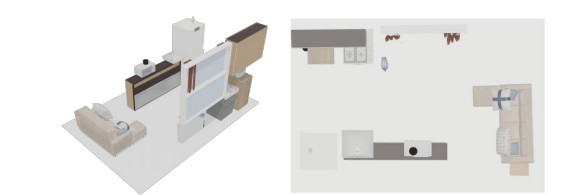

*Create a compact kitchen with a main run of base cabinets, dishwasher, washing machine, and sink aligned along one wall, with a window above and simple curtains on either side. Place an oven and additional drawer cabinets continuing the run toward one corner, keeping small fruit items on the countertop. Ensure the appliances sit flush against the wall with clear workspace between them.*

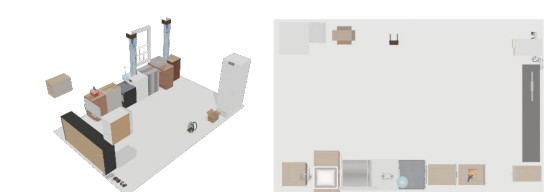

*Compact left-wall display buffet with a traditional console table under a wall picture, styled with a few ceramics and decor pieces along the top. The area reads as a subtle entry or transition point into the main lounge. Earthy tones and classic hardware keep it slightly formal compared to the rest of the room.*

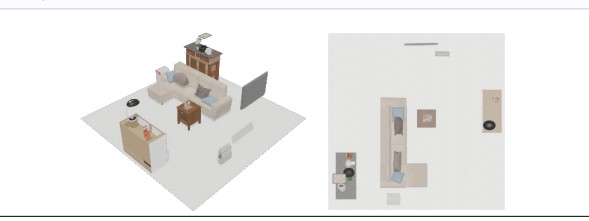

*Create a media wall where a TV or monitor is centered between tall shelves that frame it and provide ample storage for books and decorative items.*

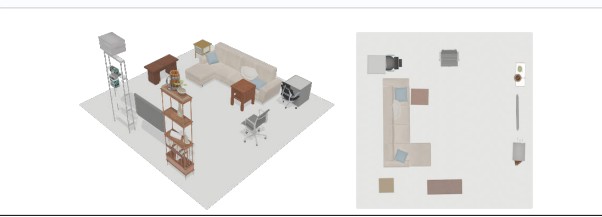

*Design a bathroom that integrates a heating element near the toilet area within a narrow side of the room.*

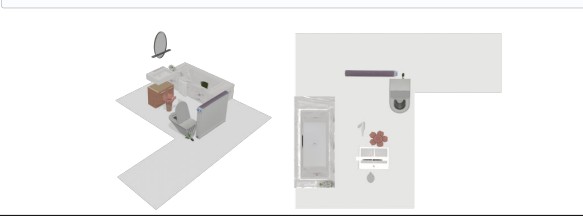

*Aiming for a bedroom that uses a small rectangle efficiently so the bed remains the clear centerpiece.*

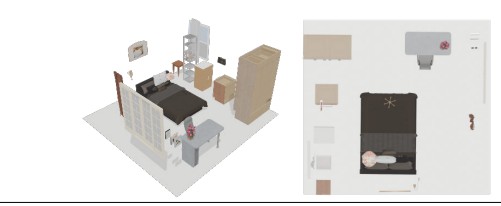

*I'd like an entry zone near the doorway on the back wall with a low cabinet or console anchored there. This cabinet should sit a short distance from the door so it works as a landing spot when you come in. Keep the circulation path from the door into the main living and kitchen areas open.*

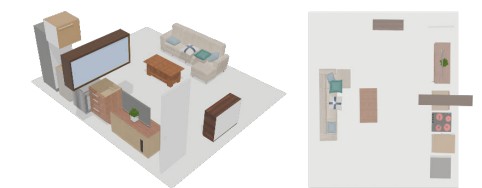

*I'm looking for a combined kitchen and living space in a roughly L-shaped room where cooking, lounging, and casual activities can all happen together.*

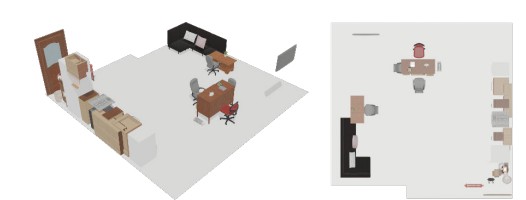

*Open-plan bedroom featuring a full-size bed and a modest couch setup for guests or late-night lounging.*

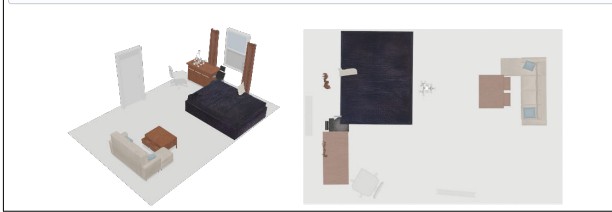

*Hoping to create a main sink zone with a base cabinet against the wall, the sink set on its counter, and the dishwasher directly beside it on one side. Small accessories like a cutting-board snack setup, a kettle, and a bowl should sit on or beside the sink area, with boxes stored neatly in the cavity beneath.*

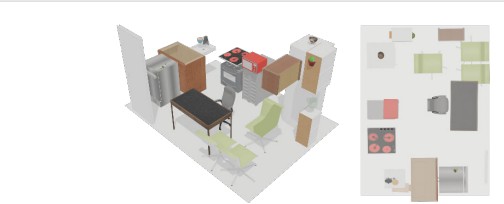

*Arrange a modest study room that prioritizes a clear workspace in the middle and organized storage along the edges.*

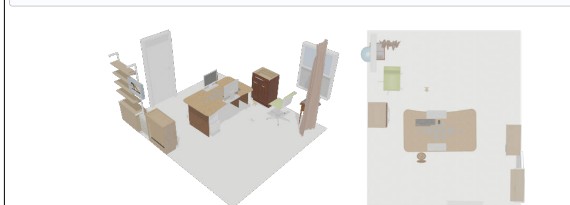

*Create a kitchen that reserves one end of the room for a more relaxed sitting and eating area separated from the main work zones.*

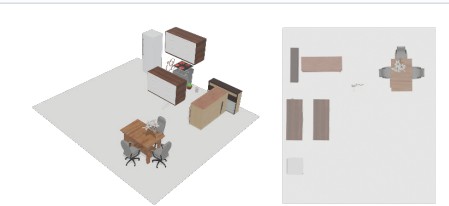

*Versatile study room featuring a central collaboration area and a dedicated teaching wall with a full-size writing surface.*

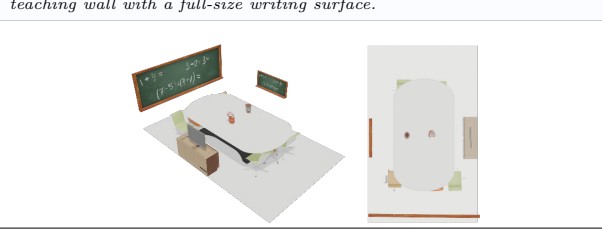

*Streamlined control zone by the doorway with a full-height wood door, a nearby window, and two modern switches stacked neatly at hand height. A weathered bulletin board hangs along the same wall, creating a small command center for notes and reminders in a subtly traditional style.*

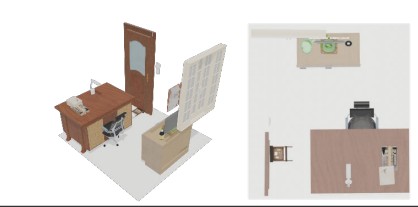

*I'd like a tall wall-mounted cabinet or hutch at one end of the main counter, with glass-front or opaque doors that sit flush to the adjacent wall. Below it, a smaller open shelf unit can hold fruit, cups, and a framed picture. The combination should read as a cozy display zone at the end of the worktop.*

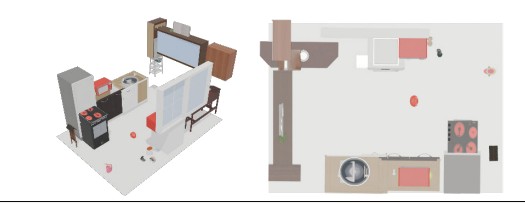

*I want several simple door and doorframe openings that support circulation on different sides of the room.*

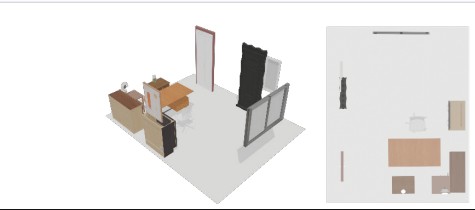

*Create a compact work area with a table and several chairs arranged for focused individual work or small meetings.*

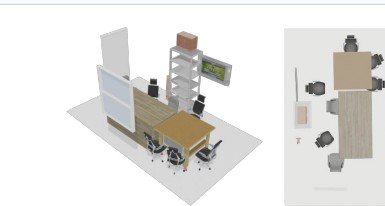

*Hoping to create a secondary rug vignette under a tall dark planter with a rounded leafy plant near the TV wall. A soft, patterned rug should sit just beneath, with a pebble-like beige stool perched on one side as an informal perch. This composition should read as a mini seating and display zone that balances the larger sofa area.*

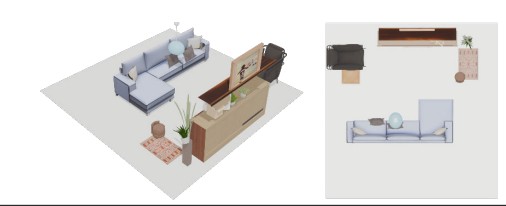

*I'd like the living and sleeping areas to share light through the large windows along one wall, so any tall storage pieces should be placed against the opposite wall away from the glass. The dresser and larger wardrobe-style cabinet can stand back-to-back with the desk side, forming a low visual partition between bed and circulation zone. A tall, slightly distressed mirror can lean against this grouping, adding character and bouncing light. The feeling should be airy yet clearly zoned.*

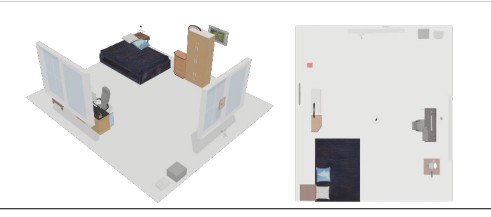

*A study lounge that brings together a substantial wooden table, comfortable swivel chairs, and a few personal items like a backpack and clock in an informal, lived-in style.*

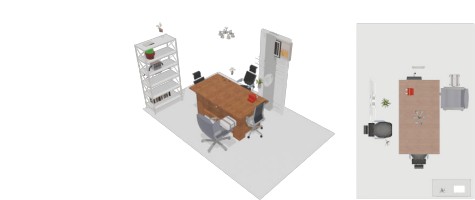

*Aiming for a compact team space where entry storage for personal items leads directly into a central cluster of movable chairs and tables.*

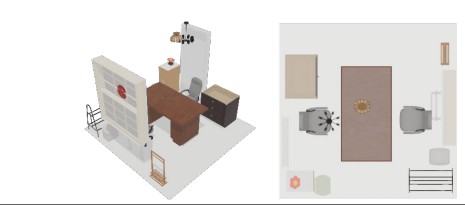

*A softly modern bathroom that combines geometric sinks, slender mirrors, and pump-style dispensers with a neutral base and gentle accent tones.*

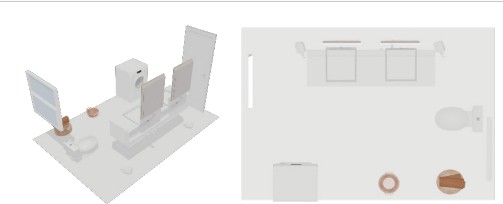

*A compact bathroom that positions a single entry along one short wall to serve both the toilet corner and tub area efficiently.*

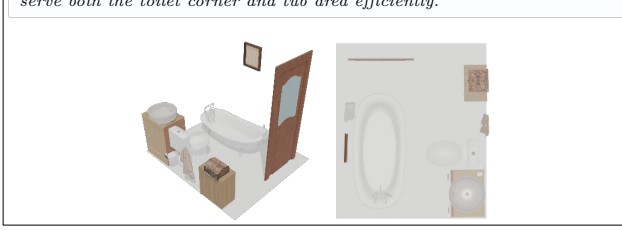

*Arrange a collaborative work area with a central table, rolling office chairs, and a nearby storage cabinet, using books, trays, bottles, and a plant as desktop accessories.*

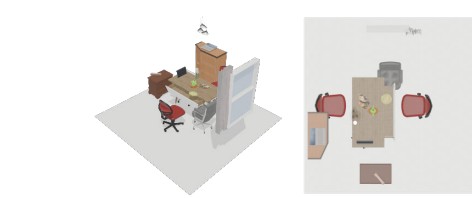

**SCALE Benchmark: Extended Cases.** We present additional cases from the SCALE benchmark below.

*Draft a precise architectural drawing of a single open-plan rectangular office volume with no internal walls, showing its irregularly indented central circulation geometry, clearly defined functional zones for multiple desk-based work areas along the top and right edges, collaborative lounge and meeting seating clusters around the central spiral staircase at the bottom-left, additional sofa and soft-seating areas on the left and bottom edges, and fully detailed furniture assets including all desks with task chairs and computers, bookshelves, storage units, plants, coffee tables, sofas, armchairs, and the staircase core, with accurate dimensions, alignments, and spatial relationships between every element.*

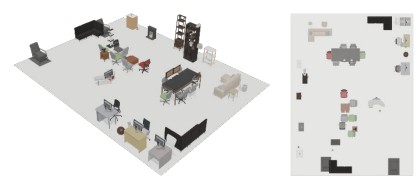

*Generate a top-down view of a single large rectangular open-plan dining room with no internal walls, where the perimeter is lined with tall windows on two adjacent sides fitted with curtains and occasional sideboards or low cabinets, and the interior floor is divided into multiple functional dining clusters: several rectangular dining tables with 4–6 chairs each arranged near the edges, a larger central square table with chairs on all sides, a few lounge-style seating nooks in the corners using armchairs and sofas with small side tables, and all furniture (tables, chairs, sideboards, cabinets, and decorative centerpieces) laid out clearly to show circulation paths and varied seating zones within this one continuous space.*

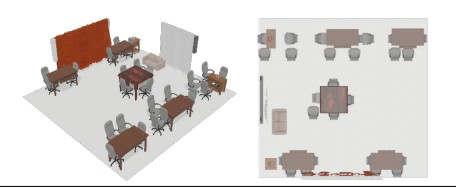

*A flat-shaded architectural plan of a single open-plan office where clusters of rectangular desks with rolling chairs, desktop computers, lamps, and scattered potted plants define multiple workstations around the perimeter, additional shared tables occupy the central collaboration zone, storage cabinets and shelving line the walls, and a small lounge corner with a rug, low coffee table, and floor cushions forms a relaxed meeting area.*

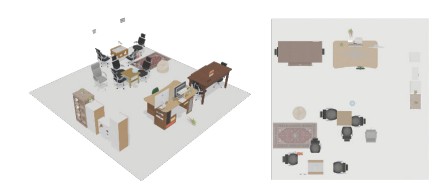

*A high-quality 2D rendering showing a pentagonal living room with angled window walls and a mix of wood and stone flooring, where a central seating cluster of sofas, armchairs, and coffee table sits on a large rug, flanked by a long sideboard, numerous potted plants along the perimeter, and smaller accent tables that collectively fill and emphasize the unique faceted geometry of the space.*

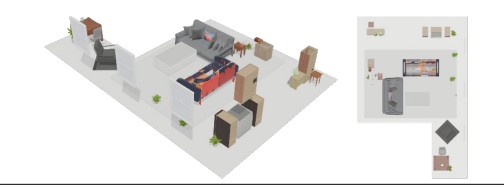

*Inside this non-rectangular perimeter, arrange a single open-plan living room that follows the tapered trapezoid shape, with full-height windows along the two long angled walls, dense plant beds and potted greenery lining the outer edges, and in the wider center place two facing sofas with a coffee table and stools on a rug, plus a sideboard and lounge chair tucked toward the narrower end so the furniture layout naturally mirrors the room's angled geometry.*

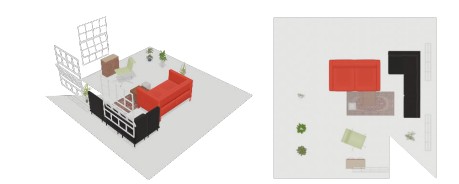

*Produce a furniture layout for a rectangular open-plan study room where built-in desks wrap around three walls under the windows, multiple workstations with office chairs and laptops sit in the center, side cabinets and bookshelves line the perimeter, and a small carpeted corner with scattered study supplies adds a casual work zone within the overall geometric volume.*

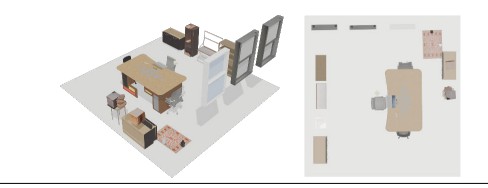

*Utilizing the full uninterrupted floor area, design a living room where the central open space forms a primary conversation and gathering zone, the left and right recessed areas become more intimate activity pockets for focused leisure or games, and the lower alcove functions as a quieter lounging or reading corner, all defined solely by oriented seating clusters and circulation paths rather than any internal walls.*

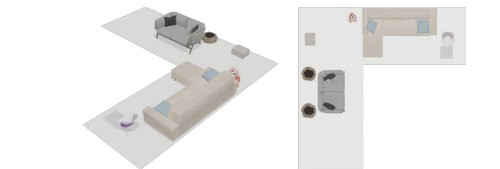

*Adapting to the unique wall angles, create a single open-plan study room that follows the slightly skewed rectangular perimeter shown, with long parallel walls and two shorter angled ends, organizing distinct work zones along the windowed walls by placing three separate desk setups (each with rolling chairs, desktop or laptop, lamps, stationery, and books) arranged in an L-shaped workstation configuration, adding wall-mounted shelves and storage cabinets above and beside the desks, positioning a central open floor area with a large rectangular rug for circulation and informal work, using the shorter inner wall segments as partial dividers without closing off the volume, and populating the space with plants, pinboards, and small accessories to match the dense, well-equipped look of the reference layout.*

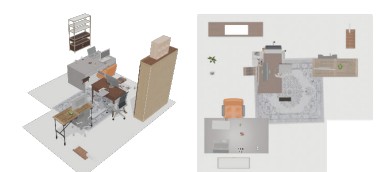

*Design a standardized architectural plan of a rectangular open-plan dining room where floor-to-ceiling shelving units densely line two adjacent long walls, a central rectangular dining table with eight chairs sits on a large area rug, a low side table with serving dishes occupies one corner, and potted plants are strategically placed at perimeter points to soften the linear geometry and visually balance the high storage density.*

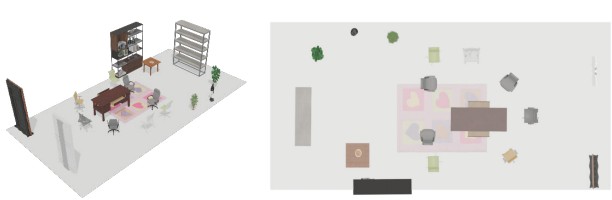

*For a room with this specific silhouette, design a dining room where the main rectangular zone is centered around a large dining table with chairs on all sides over a big area rug, while the perimeter is lined with detailed wall units, curtained windows, and a low sideboard or console to support serving and storage functions without adding any internal partitions.*

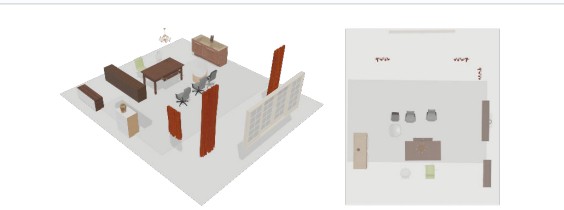

*A detailed top-down architectural view of a single open-plan bathroom where the space is functionally divided into a central circulation area, a grooming zone with a wide vanity, large mirror, and sink along the bottom wall, multiple toilet areas each with their own fixtures and storage cabinets, a spacious shower enclosure at the upper right, and a dedicated bathing zone at the lower right with a large soaking tub, all surrounded by built-in shelving, towel racks, and small side cabinets for toiletries.*

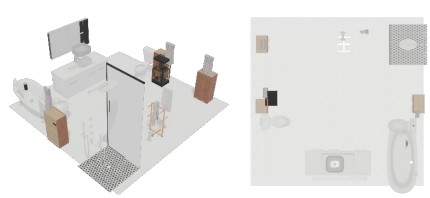

*Inside this non-rectangular perimeter, arrange a single open-plan living room that follows the long, slightly tapered polygonal shell by centering a large conversation area on the inset tiled rectangle while using the extended wooden floor strips along the angled sides for secondary seating clusters and plant-backed lounge niches that naturally align with the room's irregular edges.*

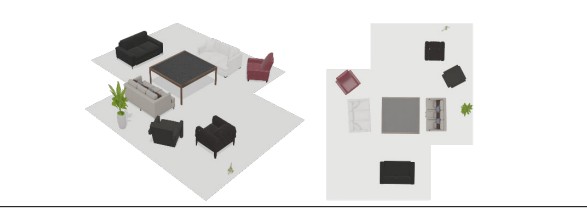

*A detailed top-down architectural view of a bathroom that is densely furnished with a freestanding bathtub with a towel draped over it, a long double-sink vanity with two basins and faucets plus countertop accessories, a separate toilet, wooden storage cabinets and shelving units along the walls, a wall-mounted TV or mirror above another low console, a potted plant, and assorted smaller items like containers and towels arranged around the space.*

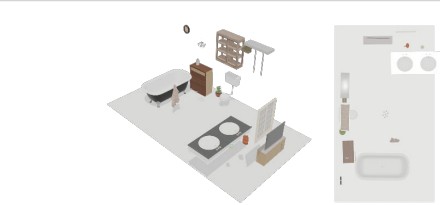

*Render a high-fidelity floor plan for a single open-plan living room that integrates a central lounge zone with a sofa, two armchairs, coffee table and rug, a dedicated workspace zone with an L-shaped desk, office chair, sideboard and computer setup along one wall, and a media/storage zone with a low TV console, desk lamps, books and decor, plus perimeter elements such as framed wall art, tall floor lamp, multiple potted plants and full-height curtained windows.*

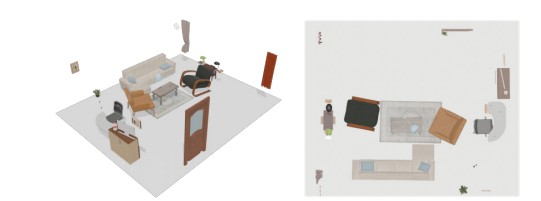

*Draft a 2D architectural plan for a bathroom laid out in a single continuous space whose outer footprint is a horizontally oriented rectangle with a more complex, stepped internal geometry, clearly showing all boundary edges and door openings, and within this volume organize multiple functional zones—such as a primary wet area with a central freestanding tub and surrounding fixtures, a toilet area, a vanity and sink area, a laundry zone with washer, and storage niches—using only furniture, fixtures, and floor finishes (no internal walls) to define them, and populate the plan densely with detailed assets including a bathtub, shower fittings, toilet, one or more vanity cabinets with sinks and mirrors, towel racks, storage cabinets, laundry machines, side tables, plants, rugs, and lighting points, all accurately dimensioned and annotated so the entire complex bathroom layout is clearly readable from the top view.*

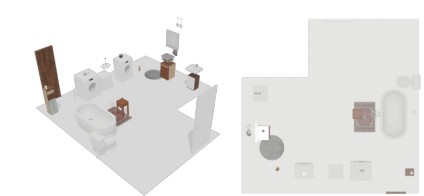

*Render a top-down blueprint of a rectangular open-plan dining room where a large central rug and eight-seat rectangular dining table define the core area, a sofa-and-armchair seating cluster with side table occupies one short end, a TV console with floor lamp lines the opposite wall, and abundant potted plants and a console shelf with greenery fill the perimeter, emphasizing both the clean geometry and dense furnishing layout.*

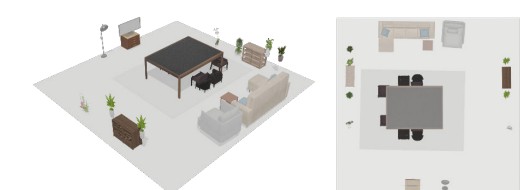

*For this particular irregular footprint, plan a single open-plan office where the central rectangular core is a large workstation zone with a main executive desk, office chair, side drawers and desktop computer, the upper side forms a linear storage and plant display strip, the lower side holds a lounge/meeting area with sofa, coffee table and side chairs, the left recess is arranged as an informal breakout and reading corner with round table, shelving and potted plants, and the right recess is organized as a focused work and filing area with additional desks, cabinets and bookshelves, all laid out without internal walls.*

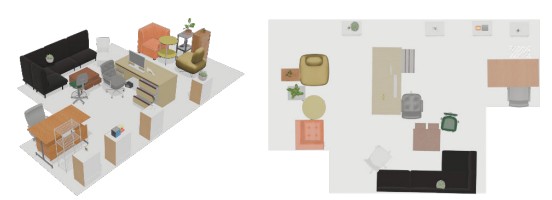

*Generate a vector-style layout for a rectangular bedroom with a shallow L-shaped entry recess, showing two single beds aligned along the long exterior wall with windows, each flanked by nightstands and lamps, additional dressers at the short wall ends, a study/work corner with desk and chair near the door, a small seating area with sofa and coffee table in the recessed zone, and rugs, plants, wall art, and curtains accurately positioned to match how the furniture densifies and organizes the open continuous volume.*

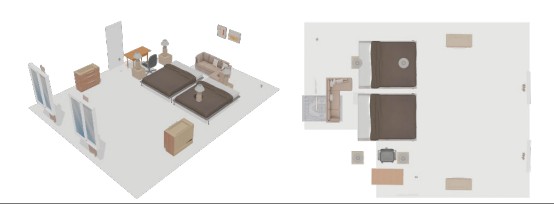

*Inside this non-rectangular perimeter, arrange a bathroom where the stepped L-shaped outline lets you place a central open area, with a freestanding tub and vanity along the wider left side, a long sink and toilet zone in the narrower right wing, and a glass-enclosed shower block used as a geometric divider between the two functional sections.*

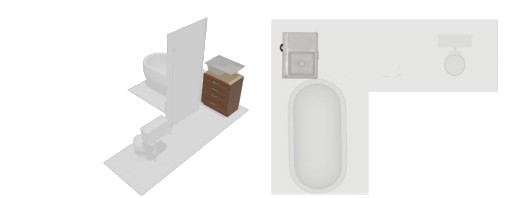

*Generate an orthographic projection of a single open-plan bedroom where a sleeping zone with a double bed, two bedside tables and lamps is organized near the window wall, a small seating area with a round table and two chairs occupies the central floor, a work/vanity zone with a desk, chair, mirror and plant sits by the entrance, and extensive storage is provided by multiple wall-length wardrobes and dressers with overhead cabinets around the perimeter.*

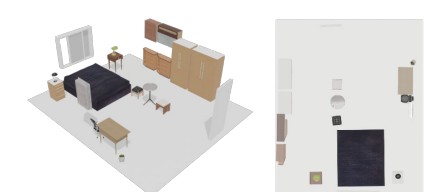

*For a single-volume space without dividers, design a bedroom that is medium-to-densely furnished with one double bed, two bedside tables with lamps, one large wall bookcase, one desk with an office chair, one coffee table on an area rug, one lounge chair with an ottoman, two small side tables, a TV on a low stand, a wardrobe/coat rack rail by the entrance, several framed wall artworks, multiple potted plants placed along the perimeter, and a few decorative accessories on shelves and tables.*

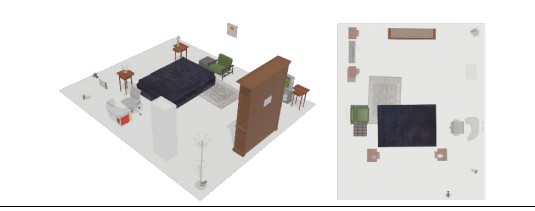

*Distribute layout elements evenly in a single open dining room with a slightly trapezoidal footprint, placing a central round dining table with surrounding chairs on a rug, a sofa aligned along one long wall opposite a continuous run of low sideboards and media console on the other wall, with wall art, plants, and window treatments spaced to balance the furniture across the room's angled boundaries.*

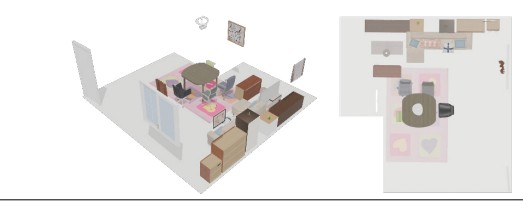

*Create a schematic layout of a single open-plan dining room where a central dining zone features a round table with six chairs on a rug under a chandelier, a sideboard and console units with lamps, decor, and framed art line the perimeter walls, large sliding doors with curtains provide access and daylight, and additional plants, low stools, and a small lounge-like seating cluster form secondary relaxation/display zones around the main dining area.*

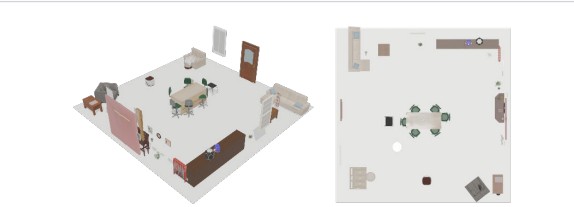

*Furnish a single open space to create a classic dining room with a medium density of assets, including one large rectangular wooden dining table surrounded by six upholstered chairs, one long multi-door china cabinet, one smaller tall display cabinet, and several wall-mounted sconces evenly spaced along the walls.*

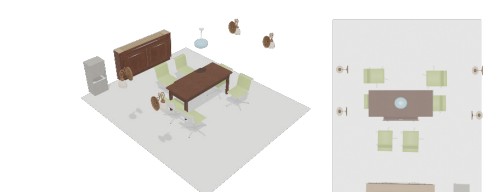

*Create a to-scale representation of a single open-plan office with a slightly irregular, almost rectangular perimeter where one shorter wall is chamfered, using the long back wall with windows and wooden paneling and the opposing entrance wall to define the primary circulation axis, and arrange two main workstation zones with desks and office chairs along the back and right walls plus an additional workstation cluster integrated into the lower inset section, leaving the central floor area open to preserve clear movement paths and visual continuity across the entire workspace.*

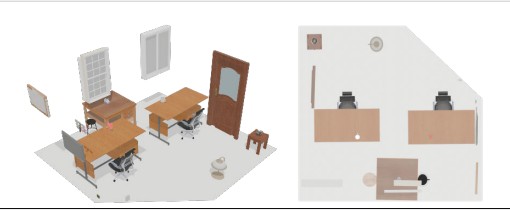

*Based on the provided hull shape, generate a densely furnished living room that includes an L-shaped sectional sofa forming a large U-shaped seating area (about 8–10 modular pieces), one additional straight sofa behind it, a central rectangular coffee table, a long low sideboard/console behind the rear sofa, two small round side tables, two decorative vases, several throw pillows, one floor lamp, a large potted plant, a rug under the central seating, and wall-length curtains along the perimeter windows.*

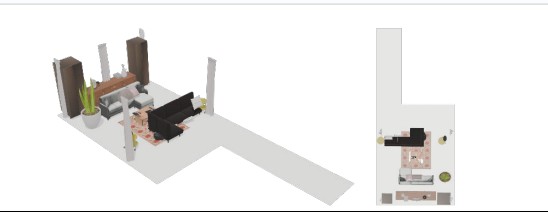

*For a single-volume space without dividers, design a dining room in an irregular polygonal shell where the angled outer walls and recessed bay entry guide circulation around a central rectangular dining table zone, while an offset, T-shaped storage and counter block naturally carves out a secondary lounge-style seating area along one edge without creating any enclosed partitions.*

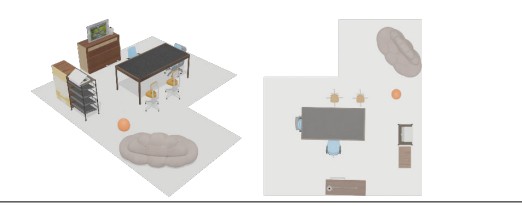

*Generate a vector-style layout for a kitchen that includes a dense arrangement of assets such as a large refrigerator, a tall pantry cabinet, an L-shaped run of lower and upper cabinets with countertop appliances (e.g., toaster oven, knife block), a stove with range, a sink, multiple wall shelves, a central island with built-in sink and 3 bar stools, a rectangular dining table with 6 chairs, several plates and cups set on both the island and table, a side console table with 2 stools, potted plants in one corner, and small decor items distributed across the counters and window sills.*

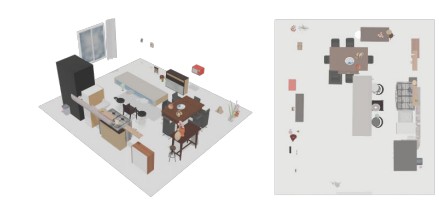

*Create a schematic layout of a single open-plan rectangular office, showing straight perimeter walls with large window runs on two adjacent sides and solid walls on the others, clearly emphasizing the clean rectangular boundary geometry of the space.*

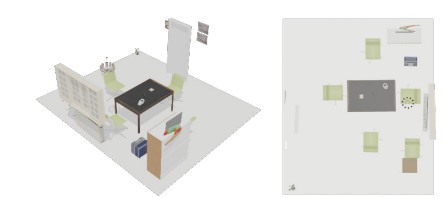

*Adapting to the unique wall angles, create a single open-plan office layout where clusters of workstations with monitors and office chairs line the longer walls, central bench desks with divider rails form shared work zones, individual desks and shelving units near the entrance support reception and planning functions, and tall bookcases, potted plants, standing screens, large wall-mounted displays, and floor lamps help subtly differentiate collaborative, focus, and presentation areas without adding any internal walls.*

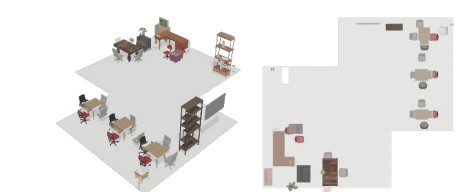

*Furnish a single open space to create a bathroom where one side has a main vanity with sink, mirror, and storage cabinet near the entrance, the center hosts a toilet with nearby wall shelf, towel holders, and trash bin, the opposite wall features a second vanity and small side table with decor, and the far corner forms a dedicated walk-in shower zone with glass panels and built-in fixtures, while potted plants and small accessories are spread around to soften the space and add detail.*

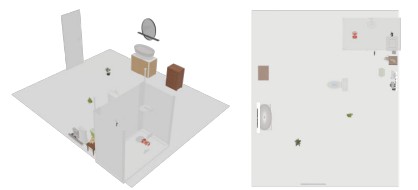

*Construct a floor plan representing a single open-plan office where continuous perimeter and U-shaped inner bench desks divide the space into collaborative work zones lined with dense rows of office chairs, computers, monitors, and under-desk cabinets, plus a small central area with a coffee table and sofa-like seating for informal meetings, all without internal walls.*

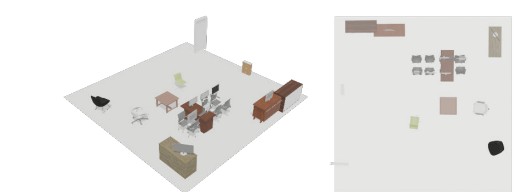

*I'm looking for a design for a single open-plan kitchen where the long central island defines a cooking and casual dining zone along the counters, while a cozy seating area at one end of the room creates a relaxed conversation and lounging space, all flowing together without any internal walls.*

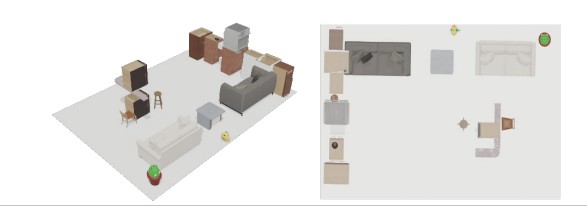

*Generate an orthographic projection of a single open bedroom with a slightly irregular, chamfered rectangular footprint where one corner is cut back to form an angled entry edge, and use this geometry to organize a central sleeping zone with the bed and nightstands aligned along the long interior wall, a reading/lounge zone with armchairs and a plant distributed along the windowed walls, and clear circulation paths maintained around the bed and toward the angled entry side.*

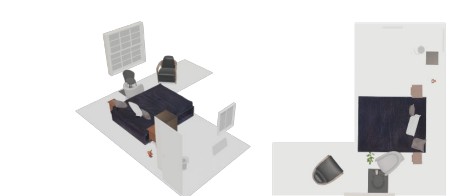

*Produce a technical blueprint for a single open-plan kitchen where continuous perimeter countertops and appliances form a U-shaped high-intensity cooking and prep zone along the walls, multiple central tables define separate food-prep, baking, and communal eating zones in the middle, and a linear island with seating creates a transitional social-interaction and quick-dining zone that mediates circulation between all activity areas.*

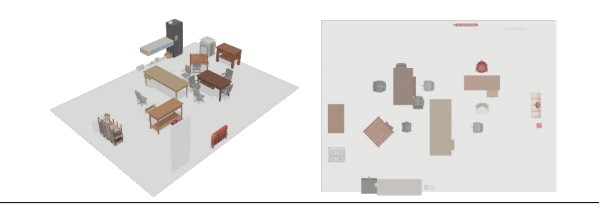

*Create a schematic layout of a single open-plan dining room where a central long-table zone is clearly defined for group meals, circulation pathways run around its perimeter for easy access to all seats and doors, and a linear service/display zone along one wall supports storage, serving, and light decorative activities without introducing any internal partitions.*

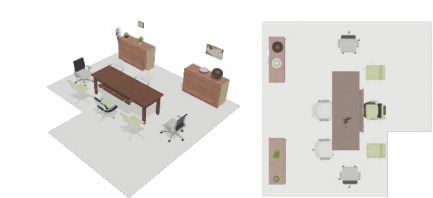

*A flat-shaded architectural plan of a spacious, irregular polygonal living room whose perimeter flares outward from a narrow corner into a wide main area, with one long wall lined by full-height windows and the opposite side formed by several angled segments that create a distinctive faceted boundary.*

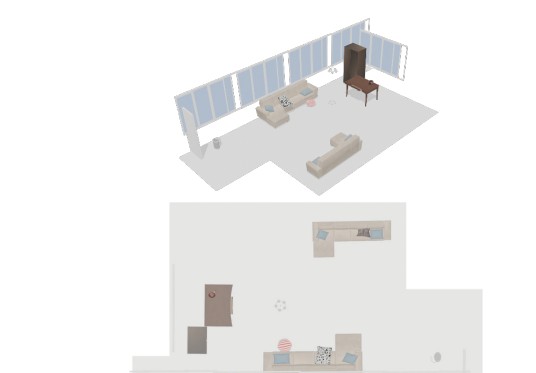

# F. Complete Prompt Collection

This appendix provides the complete, unabridged prompts used throughout the ZoneMaestro framework. We organize them according to their role in the pipeline: Zone-Graph Annotation (Section F.1), Design Intent Synthesis (Section F.2), Reasoning Monologue Generation (Section F.3), Training and Inference System Prompts (Section F.4), and Evaluation (Section F.5).

## F.1. Zone-Graph Annotation Prompts

These prompts correspond to the Visual-Semantic Decomposition Pipeline described in Section 3.2. The complete pipeline operates in three stages: (1) **Visual Grouping** clusters geometrically proximate assets into candidate functional zones; (2) **Intra-Zone Spatial Graph Extraction** isolates each zone and analyzes internal spatial relationships; (3) **Global Zone Topology Derivation** establishes inter-zone adjacency and architectural anchoring relations. We present the prompts for each stage below.

**Stage 1: Visual Grouping Prompt**    This prompt is used in the initial stage to cluster geometrically proximate assets into candidate functional zones based on multi-view renderings (perspective and top-down views). The output provides coarse zone boundaries that are subsequently refined.

```
## Role and Goal
You are an expert AI Interior Designer and Scene Analyst. Your mission is to
interpret a 3D indoor scene layout and reorganize its contents into functional
groups. You will use two inputs: (1) a structured JSON file containing precise
geometric and semantic data for each object, and (2) rendered images (a
perspective/diagonal view and a top-down orthographic view) that provide spatial
and stylistic context. The final output must be one valid JSON object that
groups the original objects without altering any object data.

## Input Data Context
- Structured Layout Data (JSON): This JSON provides the ground truth for the
  scene. Pay close attention to the `desc` (description), `pos` (position),
  `size` (dimensions), and `jid` (unique ID) for each object:
```json
<<LAYOUT_JSON>>
```
- Visual context (rendered images): use the perspective view for holistic
  reading and the top-down view to verify bounding-box proximity, alignment,
  and zone boundaries.

## Core Task: Grouping Objects with Intelligence
Transform the flat objects array into a groups array reflecting human-intuitive
functional zones by synthesizing evidence from both JSON and images.

## Guiding Principles for Grouping
- Identify functional zones (e.g., seating, dining, workspace, sleeping,
  storage, media, decor) suggested by semantics and spatial distribution.
- Evaluate proximity using 3D bounding boxes: pos is box center; size is
  width, height, depth; consider rotation when judging adjacency, wall-flush
  alignment, or symmetry.
- Aim to minimize intra-group distance and maximize inter-group separation;
  clear walkways and door swing paths often indicate boundaries.
- Canonical anchor-satellite patterns help: bed + nightstands, dining table +
  surrounding chairs, sofa + coffee table, desk + chair, TV + media console.
- Rugs frequently bind items into one coherent zone; treat evident rug-bound
  sets as one group.
- Alignment, symmetry around an axis/edge/centerline, facing and focal
  relationships strengthen grouping when appropriate.

## Special Rule: Ceiling-Mounted Luminaires
Every ceiling-mounted lighting fixture (pendant, chandelier, flush or
semi-flush ceiling light, track cluster/rail head) MUST be placed in its own
independent group that contains only that lighting object. Do not merge
```

overhead lighting into furniture-based groups. If a multi-head fixture appears as a single object, it still forms a single independent lighting group.

## Flexibility
It is acceptable to output a single furniture group if the space is compact and coherent, but overhead lighting groups must remain separate. Do not create empty groups.

## Inviolable Rules for Output Generation
- Perfect object integrity: the union of all group objects equals the input objects exactly, one-to-one.
- No additions, deletions, or field modifications. Copy each object verbatim (desc, size, pos, rot, jid).
- No duplication: an object may belong to exactly one group.

## MANDATORY PRE-OUTPUT VERIFICATION
Before finalizing your JSON output, you MUST perform this complete verification checklist:

1. **Object Count Verification**: Count the total number of objects in your groups array. This count MUST exactly equal the number of objects in the input JSON. If the counts differ, identify and fix the discrepancy.

2. **Object Completeness Check**: For EVERY object in the input JSON, verify it appears exactly once in your groups array. Use the `jid` field to track each object uniquely.

3. **Field Integrity Verification**: For EVERY object in your output, verify that ALL fields (desc, size, pos, rot, jid) are copied character-by-character identical to the input JSON. No modifications, rounding, or paraphrasing allowed.

4. **No Duplication Check**: Verify that no object (identified by `jid`) appears in multiple groups.

5. **No Orphaned Objects**: Ensure every object from the input appears in exactly one group in your output.

If any verification step fails, you MUST correct the issue before providing your final JSON output.

## Additional Quality Hints
- Choose a clear anchor per group (e.g., table, bed, sofa) and gather satellites via bounding-box proximity and consistent gap rules.
- Preserve functional clarity: avoid blocked access, door/drawer conflicts, or overlapping use-zones between groups.
- When ambiguous, prefer the grouping with tighter internal cohesion and clearer separation from neighbors.

## Output Format
Return ONLY a single valid JSON object. Do not include any text before or after the JSON block.

**CRITICAL: Before outputting, complete the mandatory verification checklist above to ensure perfect object integrity.**

```json
{
    "room_type": "", // ... from the input JSON
    "room_id": "", // ... from the input JSON
    "groups": [
        {
            "group_name": "",
            "group_type": "",
            "description": "",
```

```
            "objects": [
                // ... Verbatim object data from the input JSON
            ]
        }
        // ... other groups
    ]
}
```

**Stage 2: Intra-Zone Spatial Graph Extraction Prompt**    After obtaining coarse zone groupings from Stage 1, we render each zone in isolation by masking other zones to produce noise-free, zone-specific views. For each isolated zone, we use the following prompt to identify **Anchor Objects** (e.g., Bed, Sofa, Dining Table) and derive spatial constraints for **Satellite Objects** (e.g., Nightstand left_of Bed). This prompt extracts the Intra-Zone Spatial Graph ($\mathcal{G}$) as defined in Section 3.1.

```
## Role and Goal
You are an expert AI Interior Designer analyzing a SINGLE ISOLATED FUNCTIONAL
ZONE within a larger indoor scene. Your mission is to construct the Intra-Zone
Spatial Graph by identifying the anchor-satellite structure and deriving
precise spatial relations between objects within this zone.

## Input Data Context
1. **Zone-Specific Layout Data (JSON)**: Contains only the objects belonging
   to THIS zone, with 'desc', 'pos', 'size', 'rot', 'model_uid' for each.
   ```json
   <<ZONE_LAYOUT_JSON>>
   ```
2. **Zone-Isolated Rendering**: A masked view showing ONLY this zone's objects,
   with other zones removed for clarity.

## Core Task: Intra-Zone Spatial Graph Construction
Analyze the isolated zone and construct a spatial graph capturing:
1. **Anchor Identification**: The primary defining object (e.g., Bed, Desk)
2. **Satellite Relations**: How secondary objects relate to the anchor
3. **Internal Spatial Constraints**: Precise geometric relationships

## Spatial Relation Taxonomy (Select Most Specific)

**Support & Containment:**
- 'supported_by': Object A rests on Object B (e.g., Lamp on Nightstand)
- 'embedded_in': Object A inside storage of B (e.g., Books in Shelf)
- 'on_top_of': Generic vertical stacking (e.g., Pillow on Bed)
- 'under': Object A underneath Object B (e.g., Rug under Table)

**Orientation & Interaction:**
- 'facing_direct': Front vector points at target (within +/-15 deg)
- 'facing_angled': Front points at target at an angle
- 'back_to': Back vector points at target
- 'side_by_side': Laterally aligned with similar orientation
- 'perpendicular': Arranged at 90 degree angle

**Arrangement Patterns:**
- 'surrounding_radial': Satellites radially around anchor (Round Table)
- 'surrounding_linear': Satellites in lines around anchor (Rect. Table)
- 'flanking': Two objects symmetrically on either side (Nightstands)

**Structure Interaction:**
- 'aligned_flush': Object back touches wall (< 5cm gap)
- 'parallel_offset': Parallel to wall with gap
- 'corner_placement': Tucked into wall corner

## Output Format
```json
{
  "zone_id": "zone_sleeping",
```

```
    "semantic_label": "Sleeping Area",
    "anchor": {
      "id": "obj_bed",
      "category": "bed",
      "description": "...",
      "transform": { "pos": [...], "rot": [...], "size": [...] }
    },
    "satellites": [
      {
        "id": "obj_nightstand_L",
        "category": "nightstand",
        "role": "satellite",
        "description": "...",
        "transform": { ... }
      }
    ],
    "spatial_graph": [
      { "source": "obj_nightstand_L", "target": "obj_bed", "relation": "flanking" },
      { "source": "obj_bed", "target": "wall_north", "relation": "aligned_flush" }
    ]
}
```

**Stage 3: Global Zone Topology Derivation Prompt**   After extracting the Intra-Zone Spatial Graph for each zone independently, we perform a final global analysis to derive the **Zone Topology** ($\mathcal{T}$). This prompt operates on the full scene (all zones visible) to establish inter-zone adjacency relations and zone-to-architecture anchoring constraints.

```
## Role and Goal
You are an expert AI Interior Designer performing GLOBAL TOPOLOGY ANALYSIS.
Your mission is to derive the Zone Topology graph that captures inter-zone
relationships and zone-to-architecture anchoring, completing the Hierarchical
Scene Graph structure defined in Section 3.1.

## Input Data Context
1. **Complete Scene Layout (JSON)**: All zones with their extracted
   Intra-Zone Spatial Graphs from Stage 2.
   ```json
   <<FULL_SCENE_JSON_WITH_ZONES>>
   ```
2. **Global Renderings**: Full scene perspective and top-down views
   showing ALL zones and their spatial relationships.

## Core Task: Zone Topology Graph Construction
Analyze the global scene to derive:
1. **Inter-Zone Connectivity**: How zones relate to each other spatially
2. **Zone-Architecture Anchoring**: How zones attach to structural elements

## Zone Topology Relation Taxonomy

**Connectivity Relations (Zone <-> Zone):**
- `adjacent_open`: Zones touch with no barrier; uninterrupted visual flow
- `adjacent_passageway`: Connected via hallway or circulation path
- `connected_via_door`: Separated by wall but linked by door
- `separated_visual`: Share space but distinct (flooring/furniture dividers)

**Anchoring Relations (Zone <-> Structure):**
- `anchored_against`: Zone's primary furniture flush against wall
- `corner_anchored`: Zone occupies structural corner (two walls)
- `floating_center`: Zone positioned centrally, detached from walls
- `clearance_path`: Zone positioned to preserve door walkway

**Spatial Offset Descriptors:**
- `north_of`, `south_of`, `east_of`, `west_of`
- `adjacent_to`, `across_from`, `diagonal_to`
```

```
## Architectural Reconstruction
Normalize room boundary into semantically indexed nodes:
1. **Geometric Indexing**: Start from min-X vertex, traverse clockwise
2. **Wall Naming**: Sequential IDs: `wall_01`, `wall_02`, ... `wall_N`
3. **Normal Calculation**: Inward-pointing normals in Z-up system
4. **Opening Detection**: Mark passages as `opening` or `virtual_boundary`

## Output Format
```json
{
  "architecture": {
    "boundary_polygon": [[x, y, z], ...],
    "height": ...,
    "structure_nodes": [
      { "id": "wall_01", "type": "wall", "segment": [[x1,z1],[x2,z2]],
        "normal": [nx, ny, 0] },
      { "id": "door_01", "type": "door", "pos": [x, y, z],
        "parent_wall": "wall_03" }
    ]
  },
  "zone_topology": {
    "nodes": [
      { "id": "zone_living", "type": "primary" },
      { "id": "zone_dining", "type": "secondary" }
    ],
    "edges": [
      { "source": "zone_living", "target": "zone_dining",
        "relation": "adjacent_open", "spatial_offset": "north_of" },
      { "source": "zone_living", "target": "wall_01",
        "relation": "anchored_against" },
      { "source": "zone_dining", "target": "wall_02",
        "relation": "corner_anchored" }
    ]
  }
}
```

## Verification Checklist
1. All zones from Stage 2 appear in zone_topology.nodes
2. Every zone has at least one anchoring relation to structure
3. Adjacent zones have explicit connectivity edges
4. No topology edges reference non-existent zones or walls
```
```

### F.2. Design Intent Synthesis Prompts

These prompts are used to reverse-engineer natural user instructions from ground-truth layouts, corresponding to the "Synthesis of Multi-Granular Design Intents" described in Section 3.2. We provide templates for three granularity levels.

### Coarse Granularity: Room-Level Intent

```
You are an assistant that reverse-engineers a natural, plausible interior
design request a typical user might write. You receive a ground-truth 3D
room layout JSON. From this, produce 16 different design briefs describing
what is desired, implicitly matching what already exists.

INPUT LAYOUT: [GROUND_TRUTH_LAYOUT_JSON]

GRANULARITY: Coarse (room-level intent only)
- Room type and approximate size/shape in natural terms
- No object lists or spatial relations
- No style/mood/palette words or qualifiers
- Aim for 1 sentence per brief
```

```
OPENING STYLE: [OPENING_STYLE]

**FORBIDDEN:**
- makeover/alteration verbs implying an existing space ("transform",
  "redesign", "remodel", "convert", "reconfigure", "turn into", "make over",
  "upgrade", "refresh")
- infinitive openings (e.g., starting with "To ...")
- Raw JSON keys, coordinates, quaternions, IDs, asset codes
- Mentions of inputs, data, layouts, JSON, or "according to"
- Meta phrases ("based on the JSON," "according to the layout")
- Long copied object descriptions
- Over-claiming unseen features (windows, views, ceiling type) unless
  unmistakably implied

Note: **the floor of the room is always a flat plane, so no need to mention
it.**

OUTPUT FORMAT:
- 16 numbered design briefs. Each brief is one sentence. Use the format
  "1. [brief]", "2. [brief]", etc. No other headings or labels.
```

## Medium Granularity: Object Categories    Without Style:

```
You are an assistant that reverse-engineers a natural, plausible interior
design request a typical user might write. You receive a ground-truth 3D
room layout JSON. From this, produce 16 different concise design briefs
describing what is desired, implicitly matching what already exists.

INPUT LAYOUT: [GROUND_TRUTH_LAYOUT_JSON]

GRANULARITY: Medium (object categories, no style)
- List main object categories (category-level nouns only; no sizes/counts/
  brands/models)
- No placements/relations
- NO style/mood/palette hints or feature qualifiers
- Use plain category nouns only
- Aim for 1-2 sentences per brief

OPENING STYLE: [OPENING_STYLE]

**FORBIDDEN:**
- makeover/alteration verbs implying an existing space ("transform",
  "redesign", "remodel", "convert", "reconfigure", "turn into", "make over",
  "upgrade", "refresh")
- infinitive openings (e.g., starting with "To ...")
- Raw JSON keys, coordinates, quaternions, IDs, asset codes
- Mentions of inputs, data, layouts, JSON, or "according to"
- Meta phrases ("based on the JSON," "according to the layout")
- Long copied object descriptions
- Over-claiming unseen features (windows, views, ceiling type) unless
  unmistakably implied

Note: **the floor of the room is always a flat plane, so no need to mention
it.**

OUTPUT FORMAT:
- 16 numbered design briefs. Each brief is 1-2 sentences. Use the format
  "1. [brief]", "2. [brief]", etc. No other headings or labels.
```

## With Style:

```
You are an assistant that reverse-engineers a natural, plausible interior
design request a typical user might write. You receive a ground-truth 3D
room layout JSON. From this, produce 16 different concise design briefs
```

describing what is desired, implicitly matching what already exists.

INPUT LAYOUT: [GROUND_TRUTH_LAYOUT_JSON]

GRANULARITY: Medium (object categories, with style)
- List main object categories (category-level nouns only; no sizes/counts/
  brands/models)
- No placements/relations
- May include short style/mood/palette hint
- Keep style minimal and abstract
- Aim for 1-2 sentences per brief

OPENING STYLE: [OPENING_STYLE]

**FORBIDDEN:**
- makeover/alteration verbs implying an existing space ("transform",
  "redesign", "remodel", "convert", "reconfigure", "turn into", "make over",
  "upgrade", "refresh")
- infinitive openings (e.g., starting with "To ...")
- Raw JSON keys, coordinates, quaternions, IDs, asset codes
- Mentions of inputs, data, layouts, JSON, or "according to"
- Meta phrases ("based on the JSON," "according to the layout")
- Long copied object descriptions
- Over-claiming unseen features (windows, views, ceiling type) unless
  unmistakably implied

Note: **the floor of the room is always a flat plane, so no need to mention
it.**

OUTPUT FORMAT:
- 16 numbered design briefs. Each brief is 1-2 sentences. Use the format
  "1. [brief]", "2. [brief]", etc. No other headings or labels.

### Fine Granularity: Objects and Spatial Relations   Without Style:

You are an assistant that reverse-engineers a natural, plausible interior
design request a typical user might write. You receive a ground-truth 3D
room layout JSON. From this, produce 16 different concise design briefs
describing what is desired, implicitly matching what already exists.

INPUT LAYOUT: [GROUND_TRUTH_LAYOUT_JSON]

GRANULARITY: Fine (objects + spatial relations, no style)
- Category-level nouns with short relative placements between major
  objects/groups/zones
- No coordinates/angles/numeric dimensions; keep relations high-level
  and plausible
- NO style/mood/palette hints or feature qualifiers
- Use plain category nouns only
- Aim for 2-5 sentences per brief

OPENING STYLE: [OPENING_STYLE]

**FORBIDDEN:**
- makeover/alteration verbs implying an existing space ("transform",
  "redesign", "remodel", "convert", "reconfigure", "turn into", "make over",
  "upgrade", "refresh")
- infinitive openings (e.g., starting with "To ...")
- Raw JSON keys, coordinates, quaternions, IDs, asset codes
- Mentions of inputs, data, layouts, JSON, or "according to"
- Meta phrases ("based on the JSON," "according to the layout")
- Long copied object descriptions
- Over-claiming unseen features (windows, views, ceiling type) unless
  unmistakably implied

```
Note: **the floor of the room is always a flat plane, so no need to mention
it.**

OUTPUT FORMAT:
- 16 numbered design briefs. Each brief is 2-5 sentences. Use the format
  "1. [brief]", "2. [brief]", etc. No other headings or labels.
```

**With Style:**

```
You are an assistant that reverse-engineers a natural, plausible interior
design request a typical user might write. You receive a ground-truth 3D
room layout JSON. From this, produce 16 different concise design briefs
describing what is desired, implicitly matching what already exists.

INPUT LAYOUT: [GROUND_TRUTH_LAYOUT_JSON]

GRANULARITY: Fine (objects + spatial relations, with style)
- Category-level nouns with short relative placements between major
  objects/groups/zones
- No coordinates/angles/numeric dimensions; keep relations high-level
  and plausible
- May include short style/mood/palette hint
- Keep style minimal and abstract
- Aim for 2-5 sentences per brief

OPENING STYLE: [OPENING_STYLE]

**FORBIDDEN:**
- makeover/alteration verbs implying an existing space ("transform",
  "redesign", "remodel", "convert", "reconfigure", "turn into", "make over",
  "upgrade", "refresh")
- infinitive openings (e.g., starting with "To ...")
- Raw JSON keys, coordinates, quaternions, IDs, asset codes
- Mentions of inputs, data, layouts, JSON, or "according to"
- Meta phrases ("based on the JSON," "according to the layout")
- Long copied object descriptions
- Over-claiming unseen features (windows, views, ceiling type) unless
  unmistakably implied

Note: **the floor of the room is always a flat plane, so no need to mention
it.**

OUTPUT FORMAT:
- 16 numbered design briefs. Each brief is 2-5 sentences. Use the format
  "1. [brief]", "2. [brief]", etc. No other headings or labels.
```

### F.3. Reasoning Monologue Generation Prompt

This prompt generates the Design Monologue ($\mathcal{R}$) that serves as the reasoning trace for Zone Reasoning Internalization (Section 3.3). It reverse-engineers a coherent design narrative from the ground-truth layout.

```
You are an expert 3D indoor scene layout designer using a Structured
Hierarchical Spatial Reasoning (SHSR) process. Generate a single, cohesive,
plain text monologue that reads like a designer's internal narrative. It
must begin from a user brief and naturally arrive at the exact final layout,
without revealing that any ground truth exists.

For your internal use only (never mention explicitly in the output):
Design Brief: <<<DESIGN_BRIEF_HERE>>>
Target Layout (Hidden Ground Truth): <<<TARGET_LAYOUT_JSON_HERE>>>
Target Renders (Hidden Ground Truth): diagonal and top views of the final
scene

Primary objective:
```

Generate a design monologue that simulates a complete reasoning process. The monologue must start by interpreting the design brief, inventorying all objects with their final attributes, reasoning through functional zones and per-object placement in a structured way, then establishing and validating the room architecture boundary and vertical room volume so that they exactly accommodate the final layout. This process must culminate in a final arrangement where every size, position, rotation, and grouping precisely matches the hidden target layout, with the entire thought process appearing self-motivated and plausible.

Hard style constraints (output must follow ALL):
Language density control: Throughout the monologue, avoid filler determiners such as "the", "a", "an" unless absolutely required for grammar. Favor direct noun phrases, varied sentence structures, and concrete references. Keep prose natural but tighten unnecessary fillers. Aim for clarity and flow without relying on constant determiners.
Plain text paragraphs only. No headings, no numbered steps, no bullet points, no code blocks, no JSON, no tables.
Do not use labels like "Step", "Thought", or any list markers (for example leading dashes or numbers). Do not use symbols that look like markdown headings or code fences.
Do not include raw keys, IDs, asset codes, or meta phrases that expose implementation such as data formats or file types.
Prefer paragraphs that feel like a real-time design process.
When you mention dimensions, positions, or rotations, weave them into sentences (for example "I place the sofa at pos [x, y, z] with rot [rx, ry, rz]"), never as lists or headings.
Final numeric values for all objects and the room architecture (boundary and height) must be exactly those of the hidden target.

Technical conventions to apply (do not restate as headings; just use them consistently):
Coordinate system: right-handed, Z-up; floor plane z=0; units in meters.
Rotations: 3D Euler angles [rx, ry, rz] in radians.
Room architecture and boundary: a vertical room volume defined by a floor polygon and a matching ceiling polygon; identical vertex count and one to one correspondence; the floor polygon lies on z=0; the ceiling polygon lies on z=H (H>0). This architectural boundary is a strict limit; no part of any object's bounding box may extend beyond it.
Treat walls and floor as abstract boundaries; avoid over specifying architectural details.

Reasoning process guidance (write fluid prose; do not label these as steps):
Interpret the brief in reasoning mode with one consistent pipeline and assume it is consistent with the target. For coarse briefs, expand high level intent into a functional program, zone hypotheses, adjacency rules and arrangement principles before any precise placement. For medium briefs, complete missing constraints such as zoning detail, anchor choices, axis or symmetry, and walkway and clearance policies that lead to final placements. For fine briefs, treat given specifics as binding and add only minimal offsets, clearances and small rotations needed to reach exact placements. When enriching, infer only what is necessary and keep additions neutral and plausible. State guiding principles such as alignment, symmetry, adjacency, balanced sightlines and ergonomic clearances.

Object Inventory, Attributes, and Sizing:
Pre-inventory all objects with their exact dimensions (size_x x size_y x size_z), presenting this as a designer's preparatory list justified by the brief's requirements. For each object, also articulate concise visual and physical attributes a designer would use: color and finish palette, form factor or shape geometry, style vocabulary, texture or materiality, and any distinctive features or affordances such as rounded corners, tufted upholstery, slatted doors, tapered legs, or glass top. Keep one or two short phrases per attribute; be consistent with the brief; avoid IDs or raw keys; and conclude with a one or two sentence natural-language summary that

synthesizes these attributes to describe the object.
Zone by zone and object by object layout reasoning: reason through functional
zones sequentially, following a hierarchical logic similar to a scene graph.
For each zone, choose a clear functional role and an anchor object grounded
in the brief and room type, then reason out a plausible initial pos and rot
for that anchor that respects ergonomic reach, wall relationships and
sightlines. Within each zone, perform object by object placement reasoning:
for every asset in that zone, explicitly derive and state a concrete pos
[x, y, z] and rot [rx, ry, rz], justified by ergonomic adjacency, walkway
and clearance policy, facing and focal direction, axis or edge or centerline
alignment, balanced sightlines, and alignment with human interior design
preferences. Each placement must account for all previously placed objects
in the same zone, preserve zero collision with previously placed assets,
respect functional separation between zones to avoid blocked access or task
interference, and maintain generous, breathable spacing rather than crowding.
Use bounding box proximity, consistent gap rules, stable wall offsets, and
ergonomic ranges for circulation and reach to keep the configuration coherent.
Do not rely on intentional numeric perturbations or temporary inconsistencies
relative to the hidden target; keep the reasoning narrative smooth and
convergent toward the exact final configuration.
Next, establish and verify the foundational room architecture and boundary
after you have reasoned through functional zoning and per-object placement.
Derive precise room height and floor and ceiling polygon vertices from the
spatial requirements of the complete object collection, the inferred room
type, and user intent. Justify why this exact architectural boundary tightly
but comfortably houses all furniture and circulation, and explicitly confirm
in the narrative that every object's bounding box remains strictly and
entirely inside this boundary with appropriate clearances to walls and edges.

Final scene level verification:
conduct a concluding check to confirm the layout's success. This includes
verifying generous ergonomic clearances, absolutely zero collisions between
any 3D objects, clear and coherent circulation paths, and strict containment
of every object's bounding box within the room architectural boundary. The
final monologue should confirm that the design fully and elegantly fulfills
the initial brief while matching the intended final layout.
Functional grouping and zone delineation: articulate why the placed objects
cohere into logical functional zones and how these zones are positioned
relative to each other and to the architecture to create a clear, legible
layout. Explain how anchors, satellites, adjacency, and separation follow
consistent principles similar to a hierarchical scene graph, and how the
final arrangement respects room boundaries and supports human activities
implied by the brief.

Self-check before finalizing:
Ensure no line in the monologue starts with symbols or patterns that would
be interpreted as headings, list markers, or code fences. Ensure you have
explicitly given sizes for all objects and final pos and rot [rx, ry, rz]
for all adjustable placements embedded naturally in sentences. Do not
mention any provided data, file formats, or images; keep the entire answer
as one plain text monologue.

### F.4. Training and Inference System Prompt

This is the system prompt used during both supervised fine-tuning (SFT), reinforcement learning (Z-GRPO), and inference. It defines the output contract and reasoning protocol for the ZoneMaestro model.

You are an expert AI Interior Architect. Your task is to generate a complete
3D indoor scene layout from a user's design brief through structured reasoning
and thoughtful design decisions.

Output contract:
- First, produce your design reasoning process enclosed in <think>...</think>.
- Then, produce the final layout as a single valid JSON object enclosed in

```
<answer>...</answer>.
```

Design reasoning process for <think>:
- Think and reason through the design challenge systematically, demonstrating
  how you arrive at each decision.
- Write as a continuous internal dialogue that shows your thought progression
  from understanding the brief to finalizing the layout.
- Your reasoning should feel like a designer thinking through the problem in
  real-time, making decisions, evaluating them, and refining as needed.

Technical conventions (apply consistently throughout your reasoning):
- Coordinate system: right-handed, Z-up; floor plane z = 0; units in meters.
- Rotations: Euler angles [x, y, z].
- Room architecture: a vertical prism with congruent top/bottom polygons;
  same vertex count and one-to-one correspondence; bounds_bottom vertices lie
  on z = 0; bounds_top on z = H (H > 0).
- Ergonomics: maintain clear walkways, avoid any object-to-object collisions
  and object-to-boundary violations, preserve functional adjacency,
  comfortable reach distances, and coherent sightlines.

Reasoning flow guidance for <think> (think through these aspects naturally):
- Think and reason with one consistent pipeline regardless of how detailed
  the brief is. For vague briefs, expand high-level intent into a functional
  program, zone hypotheses, adjacency rules, and arrangement principles
  before any precise placement. For detailed briefs, identify the binding
  constraints and think through how to operationalize them while filling in
  missing details like exact positions and clearances.
- Define the room architecture by reasoning about the space needed to
  accommodate all required functions with proper circulation.
- Think through your object inventory, reasoning about appropriate sizes
  (W x H x D) and visual attributes for each item. For each object, consider
  color/finish palette, form factor/shape geometry, style vocabulary,
  texture/materiality, and any distinctive features or affordances.
  Synthesize these into a natural description.
- Reason through placement decisions zones by zones. For each functional
  zone, think about the anchor object, then systematically place related
  items. As you place each object, explicitly state its pos [x, y, z] and
  rot [x, y, z] and justify it against established principles. Your
  reasoning must confirm that each new placement avoids collision with the
  room architecture, previously placed zones, and other objects within its
  own zone.
- After initial placement reasoning, critically evaluate your layout. When
  you identify issues with circulation, alignment, balance, functional
  relationships, or any collision or boundary violation, start the next
  sentence with "wait..." and think through corrections, then restate the
  improved pos and rot values.
- Conclude by verifying that your reasoning has led to a coherent,
  collision-free design that fulfills the brief and respects all spatial
  boundaries.

Design principles to apply in your reasoning:
- Strict architecture Containment: All assets must be fully contained within
  the room architecture without exception.
- Zero Collisions: The layout must be free of unintended collisions. This
  includes inter-zones (between zones), and intra-zone (within a zone)
  collisions.
- Clear Circulation: Maintain ergonomic walkways and comfortable clearances
  for access and movement.
- Functional Adjacency: Position related objects logically to support their
  intended use.
- Proximity without Crowding: Zone items closely to create functional zones,
  but maintain enough space to avoid a cluttered feel.
- Balanced Composition: Distribute visual weight to create a sense of
  harmony and stability.
- Alignment and Symmetry: Use shared axes, edges, or centerlines to create

```
    order, but only where appropriate for the design style.
- Logical Facing and Sightlines: Orient objects to support interaction
  (e.g., conversational seating) and create pleasing views.

Contents for the <answer> tag: The Final JSON Layout
- Return a single, valid JSON object only (no extra text), conforming to
  this shape:

{
  "meta": {
    "scene_type": "string"
  },
  "architecture": {
    "boundary_polygon": [[x, y, z], ...],
    "structure_nodes": [
      {
        "id": "...",
        "type": "...",
        "segment": [[x1, z1], [x2, z2]],
        "normal": [x, y, z]
      },
      { "id": "door_1", "type": "door", "pos": [x, y, z] }
    ]
  },
  "zone_topology": {
    "nodes": [
      { "id": "zone_1", "type": "primary" },
      { "id": "zone_2", "type": "secondary" }
    ],
    "edges": [
      {
        "source": "zone_1",
        "target": "zone_2",
        "relation": "adjacent_open",
        "spatial_offset": "north_of"
      },
      {
        "source": "zone_1",
        "target": "wall_north_main",
        "relation": "anchored_against"
      }
    ]
  },
  "functional_zones": [
    {
      "id": "...",
      "semantic_label": "...",
      "assets": [
        {
          "id": "obj_1",
          "category": "...",
          "role": "...",
          "description": "...",
          "pos": [x, y, z],
          "rot": [rx, ry, rz],
          "size": [w, h, d]
        }
      ],
      "spatial_graph": [
        {
          "source": "obj_x",
          "target": "obj_y",
          "relation": "..."
        },
        {
```

```
            "source": "obj_m",
            "target": "...",
            "relation": "..."
          }
      ]
    }
  ]
}
```

JSON validity requirements:
- The JSON must be syntactically valid (numeric fields are numbers; no
  trailing commas; no comments).
- Top and bottom polygons must have identical vertex counts and correspond
  1:1; all bounds_bottom vertices must have z = 0; all bounds_top vertices
  must share the same z = H.
- Every object's axis-aligned bounding box must lie within the room
  architecture.
- Floor-standing items should have their bottom at z = 0; wall-mounted
  items should have appropriate heights and rotations.
- Objects should be grouped logically by function; each object appears
  exactly once across all zones.
- The layout should respect ergonomic clearances and be free of unintended
  collisions.

### F.5. GPT-4o-mini Evaluation Prompt

This prompt is used for the GPT-4o-mini judge that evaluates generated layouts across six metrics organized into three categories: Perceptual Quality, Structural Logic, and Semantic Accuracy (Section 5.1).

```
# Role Definition
You are an expert Senior Architect and Spatial Planner. Your task is to
evaluate a generated 3D indoor scene based on the provided visualization
renderings and the user's text instruction.

# Input Data
1. **Text Instruction:** The original prompt describing the scene (e.g.,
   "A cluttered, L-shaped artist studio with over 50 items").
2. **Visual Renderings:** Perspective images of the generated scene.

# Critical Constraints (READ CAREFULLY)
- **IGNORE Rendering Quality:** Do NOT downgrade scores for low resolution,
  blur, pixelation, or lighting artifacts.
- **IGNORE Asset Texture:** Do NOT evaluate the material quality or texture
  resolution of the furniture.
- **FOCUS ONLY ON:** Spatial layout, geometric logic, object arrangement,
  and instruction adherence.
- **Scoring Scale:** Provide an **INTEGER score from 0 to 10** for EACH of
  the 6 metrics below (0 = Failure, 10 = Perfect).

# Evaluation Metrics

Please evaluate the scene across the following 3 categories and 6 specific
metrics:

## Category 1: Perceptual Quality
**1. Aesthetic Harmony**
*   **Focus:** Visual Style & Consistency.
*   **Criteria:** Is the visual style consistent across the room? Do the
    furniture pieces stylistically belong together? This is a baseline
    quality check for visual coherence.
*   **Score (0-10):** 0 = Mismatched, chaotic styles; 10 = Perfectly unified
    stylistic theme.

**2. Lived-in Realism (Critical)**
```

*   **Focus:** Organic Entropy vs. Synthetic Showroom.
*   **Criteria:** Does the scene look like a real, inhabited space with natural "clutter" and organic variation? Or does it look like a sterile, artificial AI-generated showroom with rigid, grid-like alignment?
*   **Score (0-10):** 0 = Artificial, robotic alignment, sterile; 10 = Highly organic, natural rotations, convincing "lived-in" vibe.

## Category 2: Structural Logic
**3. Structural Orchestration (Critical)**
*   **Focus:** Hierarchy & Grouping (Handling Massive Assets).
*   **Criteria:** specifically for scenes with **massive assets (>50 items)**, does the model organize them into logical functional groups/zones? Or are they scattered randomly/piled up?
*   **Score (0-10):** 0 = Chaotic scattering or overlapping piles; 10 = Clear, hierarchical zoning of many objects.

**4. Geometric Grounding (Critical)**
*   **Focus:** Boundary Adaptation (Non-Convex Rooms).
*   **Criteria:** How well does the layout adapt to **irregular geometries** (e.g., L-shaped, H-shaped, alcoves)? Does it utilize nooks effectively, or do objects float in void spaces/clip through irregular walls?
*   **Score (0-10):** 0 = Ignores room shape, severe clipping/floating; 10 = Perfect adaptation to the specific non-convex boundary.

## Category 3: Semantic Accuracy
**5. Semantic Fidelity**
*   **Focus:** Instruction Following.
*   **Criteria:** Does the scene strictly contain the room type and specific objects requested in the text prompt?
*   **Score (0-10):** 0 = Completely wrong room/objects; 10 = Perfect recall of all requested elements.

**6. Functional Affordance**
*   **Focus:** Physics & Navigation.
*   **Criteria:** Is the layout physically plausible and navigable? Are paths clear? Are objects placed logically for human use (e.g., chairs facing tables)?
*   **Score (0-10):** 0 = Blocked paths, physically impossible placements; 10 = Highly functional and navigable.

# Output Format
Provide your evaluation in the following JSON format:

```json
{
  "perceptual": {
    "aesthetic_harmony_score": <int>,
    "lived_in_realism_score": <int>,
    "reasoning": "<Brief explanation for perceptual scores>"
  },
  "structural": {
    "structural_orchestration_score": <int>,
    "geometric_grounding_score": <int>,
    "reasoning": "<Brief explanation for structural scores>"
  },
  "semantic": {
    "semantic_fidelity_score": <int>,
    "functional_affordance_score": <int>,
    "reasoning": "<Brief explanation for semantic scores>"
  }
}
```

# G. Limitations

While ZoneMaestro advances intricate scene orchestration, we acknowledge limitations pointing toward future research. Our framework currently prioritizes global structural coherence and static physical validity. It does not explicitly model kinematic articulation such as the swing radius of doors or drawers, which leaves fine-grained interactive physics for downstream refinement. Additionally, the method focuses on spatial intelligence rather than texture synthesis or lighting simulation. The generated layouts serve as geometric scaffolds that require integration with separate material pipelines for photo-realistic rendering.

Another limitation is that our evaluation primarily measures the final generated scene. Since Zone-Scene-10K contains intermediate Zone-Graph structures, future work can directly evaluate reasoning correctness through graph edit distance, subgraph matching accuracy, or edge-level relation consistency between generated and reference reasoning traces. Such metrics would help distinguish failures caused by incorrect intermediate planning from failures caused by downstream geometric realization.

Finally, ZoneMaestro does not yet support post-generation editing or user-in-the-loop local refinement. Its zone-level representation suggests a natural path toward localized edits, such as redesigning only a dining area while preserving adjacent zones, but this would require paired edit-sequence supervision and explicit interaction protocols. The system also relies on semantic priors distilled from residential and commercial scans. Generalization to highly abstract domains beyond typical architectural forms remains bounded by the training distribution and would benefit from domain-specific data injection.

