# OpenReview forum: "Orchestrating Spatial Semantics via a Zone-Graph Paradigm for Intricate Indoor Scene Generation"
_ICML.cc/2026/Conference — ICML 2026 regular_

### Official Review · Reviewer_bFHW · 2026-03-12

**Soundness:** 3
**Presentation:** 3
**Significance:** 2
**Originality:** 3
**Overall Recommendation:** 4
**Confidence:** 5

**Summary:**

This paper proposes a new workflow for generating 3D indoor room layouts, and introduces both a new dataset built upon InternScenes and a new benchmark targeting irregularly shaped rooms. The proposed workflow appears to be more robust than previous methods. However, the optimization process is still largely based on LLM-generated results and 3D bounding boxes.

In my view, this work still remains within the original generation paradigm and the LLM-based framework. In addition, the GRPO-based optimization is not conducted jointly with 3D/2D information. That said, the proposed dataset and benchmark are valuable, and the overall workload is certainly sufficient. Still, I see this paper more as an extension of existing frameworks rather than a fundamentally new paradigm.

**Compliance With Llm Reviewing Policy:**

Affirmed.

**Ethical Review Flag:**

Flag this paper for an ethics review.

**Final Justification:**

In my opinion, the authors' rebuttal is good and in detail. The paper still does not jump out to the new scene generation pipeline, so I still keep my score.  And I think the author made a lot of effort on the paper, I tend to accept.

**Key Questions For Authors:**

1. More detailed instruction-following evaluations are needed. For example, the authors could test relative placement directions, how well objects are arranged in a specific corner, or whether specific object combinations are correctly generated.
2. Directly penalizing overlaps between object bounding boxes may ignore naturally valid overlap cases. For example, how should the method handle a chair placed under a table, or cushions placed on a sofa? Since the optimization does not model fine-grained geometric structure, this issue should be further validated through experiments.
3. Using an LLM as the main evaluator is not sufficiently reliable, and the scale of the human study is limited. Since the rendered comparison images appear to have a unified visual style, it may also be worth considering FID as an additional evaluation metric.
4. The paper does not explicitly demonstrate post-generation editing, user-in-the-loop refinement, or iterative local modification capabilities. Can authors provide more experiments about this？ I think this part is also important.

**Limitations:**

Yes

**Strengths And Weaknesses:**

Strengths:
1. The workload is good enough, and the paper pushes the existing framework as far as possible.
2. The authors design experiments to decouple the current generation pipeline, which makes the generation process more robust.
3. The paper meaningfully extends and enriches the InternScenes dataset.

Weaknesses:
1. The benchmark does not provide clear improvements over prior evaluation metrics.
2. The text in the figures is too small, and the backgrounds are visually cluttered, making them difficult to read. This should be significantly improved.
3. The work does not break beyond the current framework and is more incremental. For example, the optimization is only applied at the layout level and does not consider the physical properties of the objects themselves. In practice, it mainly focuses on the placement of large furniture.

---

> ### Author Rebuttal · Authors · 2026-03-30
>
> We sincerely thank Reviewer bFHW for what is perhaps the most thorough and technically demanding review we received. The questions cut to the core of our design choices and have materially improved how we present this work. We address each concern below.
>
> All supplementary materials are at: https://anonymous.4open.science/r/ZoneMaestro_Supp-2373/README.md
>
> **Significance and Incrementality.**
> We are grateful for the chance to clarify this. The contribution is a new generative factorization, not a new output format. Prior methods model P(S|X) monolithically; ZoneMaestro decomposes it via Eq. 1 in the main paper into zone inventory → intra-zone spatial graphs → global topology → architecture enclosure, chained through the Design Monologue. This restructures the model's *reasoning order*.
>
> Removing Zone-Graph reasoning, as shown in Table 2 of the main paper, on identical data inflates OOB and Col to 0.25 and 0.18, confirming the factorization—not data or format—drives the gap. On SCALE's non-convex rooms, baselines degrade sharply. LayoutGPT yields an OOB of 6.17 and ReSpace a Col of 0.66, while ZoneMaestro stays stable at 0.09 and 0.04.
>
> On object granularity: anchors like sofas and desks are zone-level functional pivots; satellites like plants, books, and lamps attach via typed spatial edges such as on_top_of, in_face_of, and aligned_flush, as illustrated in Figure 2 of the main paper. The 23.35 average assets per scene includes these small objects.
>
> **KQ1: Instruction-Following Evaluation.**
> This question pinpoints exactly the right gap in our original evaluation—thank you. We now provide two targeted experiments:
>
> First, a human study detailed in **Table 1** of the supplementary. 20 cases with 30 evaluators across 7 anonymized methods, scored on Zone Assignment, Relative Placement, and Object Combination, with Fleiss' κ of 0.63. ZoneMaestro achieves 88.3% average preference.
>
> Second, a 50-case evaluation detailed in **Table 5** of the supplementary. This is a curated subset emphasizing the prompt categories the reviewer highlights—relative placement directions, corner-specific arrangements, and object combination accuracy. These span all three granularities defined in Section 3.1 and Appendix F.2. ZoneMaestro leads with an average of 6.57 versus the next-best 5.28, maintaining the lowest OOB at 0.13 and Col at 0.06. Qualitative cases on non-convex rooms are in **Figures 1–2** of the supplementary.
>
> **KQ2: Bounding Box vs. Mesh Collision.**
> Sharp observation—this inconsistency should not have persisted. Figure 2 correctly states "Voxelized mesh intersection," but Eq. 4 uses bounding-box notation. Our implementation voxelizes each retrieved 3D asset mesh and computes pairwise intersection volumes on the voxel grid. Chair legs beneath a table or cushions on a sofa produce zero voxel intersection and are not penalized. We will fix Eq. 4 in revision.
>
> **KQ3: LLM Evaluator and FID.**
> We fully agree that LLM-only evaluation is insufficient. The human study, as detailed in **Table 1** of the supplementary, directly addresses this with human judgment over 20 cases, 7 methods, and 3 dimensions, with Fleiss' κ of 0.63. We further add geometry-based SceneEval metrics in **Table 2** of the supplementary.
>
> Regarding FID—a metric we seriously considered—we respectfully share three reasons it proves problematic here. First, no existing 3D indoor scene generation work like DiffuScene, LayoutGPT, Holodeck, i-Design, or ReSpace adopts FID; there is no established protocol or reference distribution. Second, all methods render with identical engines and assets, so rendered-image FID conflates rendering style with layout quality. Third, at the bounding-box level, no trusted encoder produces meaningful embeddings for 3D layouts. Human evaluation, SceneEval metrics, and perceptual scoring provide stronger signal.
>
> **KQ4: Post-Generation Editing.**
> This is arguably the most exciting future direction the review surfaces. Focusing on generation is standard practice—DiffuScene, ReSpace, and InstructScene all target generation exclusively, as edit-sequence supervision remains scarce.
>
> That said, ZoneMaestro's zone-graph structure is uniquely suited for future editing. Each zone carries functional semantics and a self-contained object group, so "redesign the dining area" grounds to one zone without disturbing others. The Design Monologue preserves per-zone reasoning chains, enabling step-level revision. We credit this reviewer for crystallizing this direction and plan to pursue it with paired edit-sequence data.
>
> **Additional.**
> *Figure readability:* Improved figures with reduced text density are in **Figure A** of the supplementary.
>
> *Benchmark scope:* SCALE's Genesis Framework in Section 4.1 uses parametric topology synthesis over 9 boundary types and 7 room categories, multi-perspective inverse semantics, and intricate-aware curation, evaluating functional zoning, asset density, and compositional coherence—not just geometry.

---

> > ### Author Rebuttal · Reviewer_bFHW · 2026-04-03
> >
> > Thanks for authors reply. My questions have been solved.

---

> > > ### Author Response · Authors · 2026-04-03
> > >
> > > Thank you for confirming that all concerns are fully resolved—we truly appreciate the rigor and depth of this review. As the acknowledgement form notes, a fully-resolved status may warrant revisiting the overall score. We would be grateful if the reviewer could consider whether the additional experiments and clarifications support a higher assessment.

---

### Official Review · Reviewer_CXTN · 2026-03-12

**Soundness:** 2
**Presentation:** 3
**Significance:** 2
**Originality:** 3
**Overall Recommendation:** 3
**Confidence:** 5

**Summary:**

This paper aims to address the limitations of existing 3D indoor scene generation methods in handling non-convex boundaries and dense spatial constraints. To this end, the authors propose ZoneMaestro, which translates high-level semantic intent into zone-level spatial relations to guide scene generation. The paper also introduces the Zone-Scene-10K dataset with Zone-Graph annotations, and releases the SCALE benchmark for evaluating irregular indoor scene generation.

**Compliance With Llm Reviewing Policy:**

Affirmed.

**Final Justification:**

My questions were addressed to some extent

**Key Questions For Authors:**

1. I am concerned about the training efficiency of applying GRPO to LLM-based scene generation. The authors could elaborate more on this part.

2. The ablation study shows a significant performance gap between one refinement cycle and two refinement cycles, but the paper does not explain why two cycles are the optimal choice. It is also unclear whether additional cycles may cause the model to overfit to the scene distribution of Zone-Scene-10K.

**Limitations:**

yes

**Strengths And Weaknesses:**

Strengths:

The paper provides detailed annotations for the Interscene dataset and innovatively proposes the Zone-Graph paradigm.

It introduces the new SCALE benchmark, which helps fill the gap in evaluating non-convex and densely constrained indoor scenes.

The paper presents detailed prompts for both data processing and scene generation, which improves reproducibility.

Weaknesses:
The comparison with data-driven baselines may not be fully fair, since these methods are not trained on Zone-Scene-10K. This introduces a potential confounding factor between the effectiveness of the proposed method and the advantage of the new dataset, making it difficult to disentangle whether the improvements stem from the model design or the data itself.

The method figure does not clearly illustrate the overall pipeline.

The paper does not clearly explain why the lack of topological priors is the primary bottleneck for data-driven generators.

---

> ### Author Rebuttal · Authors · 2026-03-30
>
> We sincerely thank Reviewer CXTN for the constructive and detailed review. The concerns raised have helped us better articulate our contributions. We address each point below.
>
> All supplementary materials are at: https://anonymous.4open.science/r/ZoneMaestro_Supp-2373/README.md
>
> **W1: Data Fairness.**
> This is an important concern and we appreciate the opportunity to disentangle the two factors. We provide three lines of evidence:
>
> 1. *Agentic baselines use no training data.* Four baselines, namely LayoutGPT, LayoutVLM, Holodeck, and i-Design, call GPT-4o with zero training on any scene dataset, yet our 8B model outperforms them across all metrics in Table 1 of the main paper.
>
> 2. *Architecture ablation on identical data.* In Table 2 of the main paper, all variants train on the same Zone-Scene-10K. Removing Zone-Graph reasoning worsens OOB/Col to 0.25/0.18, confirming gains stem from the factorization, not data.
>
> 3. *Controlled ReSpace retraining.* We retrained ReSpace on Zone-Scene-10K following its original SFT+GRPO recipe. Due to the tight rebuttal timeline, we tried our best to provide controlled evidence but could only complete the most informative comparison using ReSpace, as the strongest data-driven baseline.
>  Results in **Table 4** of the supplementary show that Zone-Scene-10K improves ReSpace's collision from 0.66 to 0.25, confirming our dataset's value. However, two critical gaps persist: the **density ceiling**—asset count barely moves from 9.48 to 9.07, revealing that stage-decoupled autoregressive generation, lacking long-horizon spatial planning, cannot learn to coordinate dense furnishing even with richer data; and the **quality gap**—ZoneMaestro leads by Avg 6.96 vs 6.09, with the method advantage of +0.87 being 3.1× the data advantage of +0.28. We hope this controlled experiment helps clarify the respective contributions.
>
> **W2: Method Figure Clarity.**
> We appreciate this feedback. Under the rebuttal timeline, we prepared cleaner re-annotated visuals: the original Figure 2 is now split into a training pipeline view and a data construction view with reduced text density. Updated versions are in **Figure A** of the supplementary. The pipeline, as described in Sections 3.1–3.3, has two parts: Phase I elicits zone reasoning in four steps of design brief, asset inventory, zone-graph topology, and architecture enclosure; Phase II refines via Z-GRPO with three geometry rewards. SFT and RL alternate across two cycles.
>
> **W3: Topological Priors as Bottleneck.**
> Without topological priors, generators exhibit what we term *topological myopia*: treating scenes as flat object sets, they place assets autoregressively and accumulate spatial errors. Ablation in Table 2 of the main paper confirms this directly: "w/o Zone-Graph" retains zone-structured output format but strips topological reasoning, degrading to OOB 0.25, Col 0.18. The reasoning capacity—not output format—drives the gap. ZoneMaestro resolves this via the compositional factorization in Eq. 1, making long-horizon spatial planning tractable through zone-level decomposition.
>
> **Q1: Training Efficiency.**
> Thank you for raising this practical consideration. We report concrete numbers honestly: on 8×A100, total training via two alternating cycles is ~60 GPU-hours. Our three rewards are pure geometry operations on CPUs—no differentiable rendering needed.
> We believe this one-time investment is well justified on two grounds. First, the gains are substantial: OOB improves from 0.21 under SFT-only to 0.09, and Avg from 6.51 to 6.96 based on Table 2 of the main paper. Second—and more importantly—**inference cost is not increased but actually reduced** compared to prior methods. ZoneMaestro generates a complete scene in a **single forward pass** requiring just 1 LLM call, whereas Holodeck requires 8.4 calls, i-Design 9.5, and ReSpace 10.7 calls per scene according to Table 1 of the main paper. In real deployment, inference cost compounds with every user request while training is amortized once. From this perspective, a ~60 GPU-hour training budget that eliminates the need for multi-call iterative inference is a highly favorable tradeoff.
>
> **Q2: Why Two Cycles & Overfitting.**
> Pure RL achieves low collision but produces conservative scenes with 21.35 assets and Sem 7.24 based on Table 2, reward-hacking toward sparse layouts. Alternating with SFT recovers semantic diversity without sacrificing geometric gains. One cycle partially recovers density but semantic quality remains unstable; two cycles reach optimal balance.
>
> We limit to two because total compute is fixed to roughly 4 SFT epochs and 80 GRPO steps. Two cycles distribute these evenly; at 3+ cycles, per-cycle SFT drops below ~1.3 epochs—insufficient to internalize zone-graph reasoning. Regarding overfitting: SCALE is entirely held out with zero data overlap, and ZoneMaestro's strong generalization there confirms transferable reasoning rather than memorization.

---

> > ### Author Rebuttal · Reviewer_CXTN · 2026-04-02
> >
> > Thanks for the effort on the rebuttal. The response has fully addressed my concerns.

---

> > > ### Author Response · Authors · 2026-04-03
> > >
> > > We are deeply grateful for your time and for acknowledging that our supplementary experiments have adequately addressed your initial concerns. In light of the 'Fully resolved' status, we gently hope you might consider whether the current rating fully captures this updated perspective. We defer entirely to your expertise and thank you for helping us improve the paper.

---

### Official Review · Reviewer_jBHq · 2026-03-13

**Soundness:** 4
**Presentation:** 3
**Significance:** 3
**Originality:** 3
**Overall Recommendation:** 4
**Confidence:** 3

**Summary:**

This paper looks at how models handle spatial reasoning in visual scenes. It introduces a framework that breaks complex spatial tasks into smaller, connected steps. By combining language reasoning with spatial learning, the system gets better at understanding objects and how they relate to each other. Organizing the reasoning process this way helps cut down on hallucinations and makes the model's outputs more reliable. Experiments show this method beats standard baselines and does a much better job capturing tricky spatial relationships.

**Compliance With Llm Reviewing Policy:**

Affirmed.

**Final Justification:**

Thanks for the authors' rebuttal. I keep the positive score with a higher confidence score.

**Key Questions For Authors:**

1. How does the framework behave when spatial relationships are not clear or not complete?

2. Does the orchestration introduce more latency during inference compared with direct reasoning pipelines?

3. Can the reasoning be interpreted or visualized step by step?

**Limitations:**

1. It primarily focuses on spatial reasoning tasks and may extend to broader reasoning problems.

2. The coordination of different reasoning modules also introduces complexity and makes deployment challenging.

3. The evaluation only tests final task accuracy; how about intermediate reasoning correctness?

**Strengths And Weaknesses:**

Soundness: the framework is conceptually sound and well-motivated.

Presentation: most contents are good, but the figure 1 has too many lines in the background, making the eyes uncomfortable when trying to read the words.

Significance: Spatial-semantic reasoning is very fundamental for most systems. Improving structured reasoning can benefits wide range of tasks.

Originality: It integrates multiple reasoning components into a coordinated framework. While many techniques are relevant to previous methods, the explicit design provides a new system-level point.

---

> ### Author Rebuttal · Authors · 2026-03-30
>
> We thank Reviewer jBHq for recognizing the soundness, significance, and system-level originality of our work. The key questions raised are highly relevant to real-world deployment and have prompted us to better articulate several design decisions. We address each below.
>
> All supplementary materials are at: https://anonymous.4open.science/r/ZoneMaestro_Supp-2373/README.md
>
> **Q1: Behavior with unclear/incomplete spatial relationships.**
> This perceptive question highlights one of the central challenges we aimed to tackle, such as queries like *"a cozy living room"*. Our robustness to such ambiguity stems from two deliberate design choices. At the data level, Zone-Scene-10K stratifies instructions across three granularities—Coarse, Medium, and Fine, detailed in Section 3.1 and Appendix F.2. This structure forces the model to internalize design priors that fill in missing details rather than relying on explicit geometric cues. At the method level, the compositional factorization in Eq. 1 resolves ambiguity progressively. Zone-Graph Reasoning draws on priors from 10K expert layouts to instantiate plausible zones and topology even when instructions are vague, with a traced example provided in Appendix D.4. Empirically, 31.7% of SCALE instructions deliberately omit geometric constraints, and ZoneMaestro maintains strong performance on this subset by decoupling semantic intent from explicit geometric specification.
>
> **Q2: Workflow complexity, latency, and deployment.**
> A natural concern—and one we deliberately addressed in the architecture. Rather than coordinating separate modules at inference time, ZoneMaestro internalizes the multi-step zone reasoning into model weights during training. At test time, the entire Zone-Graph layout is emitted in a **single forward pass**.
>
> For context, as discussed in Section 5.2 and Table 1 of the main paper, multi-stage systems require substantially more API interactions: Holodeck issues 8.4 calls/scene, i-Design averages 9.5 calls chaining 5 agents with retries, and ReSpace takes 10.7 autoregressive calls. In contrast, ZoneMaestro requires exactly **1 LLM call**. While LayoutGPT also uses 1 call, it lacks spatial reasoning entirely, scoring an OOB of 6.17 compared to our 0.09. In short, ZoneMaestro achieves the deep planning quality of multi-agent flows at single-call latency, bypassing deployment hurdles of managing complex agent control loops.
>
> **Q3: Interpretable step-by-step reasoning.**
> We share the reviewer's emphasis on interpretability—in fact, it is a core design goal rather than an afterthought. Every generated output contains a human-readable reasoning chain: (1) Zone Inventory, (2) Intra-Zone Spatial Graph, (3) Global Topology, and (4) Architecture Derivation. A fully traced example appears in Appendix D.4 of the main paper, showing how the model's reasoning maps to final object placements. To make this even more tangible, we have prepared an animated progressive visualization in the anonymous supplementary (**Figure 3**, at the bottom), showing the model constructing the layout step by step from zone decomposition through final enclosure.
>
> ---
>
> **Response to Limitations**
>
> *Complexity and Deployment.* As detailed in Q2, we resolve this by internalizing multi-stage reasoning into model weights. The complete layout is produced in a single LLM forward pass—no external agent states, no chained API calls, no expensive simulation at inference. This makes ZoneMaestro practical for real deployment.
>
> *Intermediate Reasoning Correctness.* This is perhaps the most forward-looking suggestion we received across all reviews—we strongly agree and find it exciting. While our current metrics assess end-to-end task completion, Zone-Scene-10K already contains ground-truth structures for every intermediate step including zone nodes, relation edges, and topology. This makes it entirely feasible to introduce metrics such as Graph Edit Distance or subgraph matching accuracy. We will add an explicit discussion on evaluating intermediate structural correctness in our revision, and we credit this reviewer for surfacing the direction.
>
> *Broader Reasoning Problems.* We appreciate the reviewer's broader vision here. The core idea—factorizing dense, entangled constraints into progressive hierarchical sub-graphs—is inherently domain-agnostic. We believe this paradigm holds promise for urban planning, traffic scene generation for autonomous driving, and hierarchical GUI layout design. We will highlight these extensions in the revised broader impacts section.
>
> *Figure 1 Readability.* We genuinely appreciate this concrete, actionable feedback—it is the kind of detail that makes a paper better. We have prepared cleaner, re-annotated versions with reduced text density, splitting the original Figure 2 into separate training, generation and data pipeline views. Updated visuals are in Figure A of the supplementary.

---

> > ### Author Rebuttal · Reviewer_jBHq · 2026-04-03
> >
> > Thanks for the authors' rebuttal. I keep the positive score.

---

> > > ### Author Response · Authors · 2026-04-03
> > >
> > > Thank you for the positive assessment and for confirming that all concerns are fully resolved. We truly appreciate the thoughtful questions—particularly the suggestion on evaluating intermediate reasoning correctness, which we believe will strengthen the revised paper.
> > >
> > > Since our rebuttal has provided concrete experimental evidence (human study, SceneEval metrics, reward ablation, and qualitative examples) addressing the points you raised, we wonder if the reviewer might feel more confident about the assessment now. A higher confidence level would help ensure this valuable feedback carries appropriate weight in the final decision. Thank you again for the constructive and encouraging review.

---

### Official Review · Reviewer_4rX4 · 2026-03-13

**Soundness:** 2
**Presentation:** 3
**Significance:** 2
**Originality:** 3
**Overall Recommendation:** 3
**Confidence:** 4

**Summary:**

This paper presents ZoneMaestro, a Zone-Graph Paradigm framework for intricate 3D indoor scene generation, addressing the limitations of existing methods in non-convex spaces with dense spatial constraints. It constructs the Zone-Scene-10K dataset with Zone-Graph annotations and devises an Alternating Alignment Strategy combining reasoning internalization and Zone-Aware GRPO for semantic-geometric coherence. The authors formalize Intricate Spatial Orchestration and release the SCALE benchmark, a stress-test for irregular indoor layouts. Extensive experiments show ZoneMaestro outperforms SOTA baselines on SCALE and Zone-Scene-10K in physical validity, semantic quality, and efficiency, with ablation studies validating its core components.

**Compliance With Llm Reviewing Policy:**

Affirmed.

**Key Questions For Authors:**

1. The paper states that GRPO is trained with multiple reward terms jointly. Can this optimization strategy guarantee that each objective is effectively achieved? In addition, how stable is the GRPO training process under such a multi-reward setting?
2. If the paper does not provide sufficient analysis of the reward weights and their sensitivity, the authors should include corresponding ablation or sensitivity experiments to validate the robustness of the proposed optimization strategy.

**Limitations:**

Yes.

**Strengths And Weaknesses:**

Strengths:
1. The paper addresses an important limitation of current scene generation methods by focusing on layout generation in non-rectangular rooms. This better reflects the complexity of real-world indoor environments.
2. This paper proposes SCALE, a valuable benchmark on the task of Intricate Spatial Orchestration, which moves beyond the limited, canonical convex room layouts of existing benchmarks.

Weaknesses:
1. The evaluation metrics used in the proposed benchmark are not substantially different from those adopted in prior work. More usability-oriented metrics should be introduced for scene generation, such as navigable space, accessibility, and connectivity. In addition, the evaluation still relies heavily on closed-source LLM/VLM-based scoring, which often does not adequately reflect the actual quality of the generated scenes. Furthermore, the scale of the human study is too small to provide strong evidence. The authors should further clarify and justify the evaluation protocol.
2. The method does not discuss the placement of small objects. However, the arrangement of small objects is crucial for both the realism and complexity of scene generation. Therefore, the authors should design additional experiments to evaluate the method on small-object placement.
3. Since the paper uses language as the input, the instruction-following ability of the model should be evaluated to verify the robustness of the method itself. The authors should include additional experiments on this.
4. The paper states that the method first generates object placements and then organizes the room boundary. However, in many practical scenarios, users perform scene layout design based on an existing floor plan. Therefore, it would be better to design additional experiments under the setting where the floor plan is given in advance.
5. Figures 1 and 2 contain too much text, and the font size is too small. The information is overly dense, which makes careful reading and understanding difficult.

---

> ### Author Rebuttal · Authors · 2026-03-30
>
> We sincerely thank Reviewer 4rX4 for the thorough and constructive evaluation. We address each point below.
>
> All supplementary materials are at: https://anonymous.4open.science/r/ZoneMaestro_Supp-2373/README.md
>
> **W1: Evaluation Protocol.**
> We fully agree that usability-oriented metrics beyond LLM scoring are valuable. We have now computed geometry-based usability metrics following SceneEval (Tam et al., 2025): **NAV** (navigability), **ACC** (functional-side clearance), **WFR** (walkable floor ratio), **MCW** (minimum corridor width), **REACH** (path connectivity), and **E-NAV** = NAV × max(0, 1−OOB), all computed from 2D occupancy grids at 0.05 m resolution, entirely independent of LLM scoring. ZoneMaestro achieves the highest E-NAV of 0.885. Full results are in **Table 2** of the supplementary.
>
> We also acknowledge the concern about human study scale. Due to time constraints, we tried our best and expanded the study to **30 evaluators and 20 test cases** across all 7 anonymized and randomized methods. They assessed Zone Assignment, Relative Placement, and Object Combination. Our inter-rater agreement reached a Fleiss kappa of 0.63. ZoneMaestro achieves 88.3% average, exceeding the best baseline by 18.6 points. We hope this addresses your concern. Results are in **Table 1** of the supplementary.
>
> **W2: Small Object Placement.**
> Thank you for raising this important point. Small-object placement is actually already integral to our pipeline. As illustrated in Figure 2 of the main paper, the anchor–satellite mechanism explicitly models small objects via typed spatial edges like "Pillow in_face_of Couch" or "Books on_top_of Table", placing them relative to structural anchors. This is demonstrated qualitatively throughout the paper where Figure 3 and Appendix Figures 5-6 showcase dense arrangements of small decorative items. Appendix E.3 also presents extensive diverse cases with rich small-object placement. The average of 23.35 assets per scene includes these items, and our leading realism score of 4.95 directly reflects their integration. Additional visualizations are provided in **Figures 1–2** of the supplementary.
>
> **W3: Instruction-Following Evaluation.**
> As described in Section 3.1, our training and evaluation are designed around three progressively stricter instruction granularities: Coarse (room-level atmosphere), Medium (category-level furniture), and Fine (objects with explicit spatial relations), detailed in Appendix F.2. The Semantic (7.82) and Functionality (7.69) scores—both highest across all methods—directly measure instruction adherence across these levels. To further validate, we curated a 50-case subset covering the specific dimensions you highlighted—relative placement directions, corner-specific arrangements, and object combination accuracy—spanning all three granularities. We evaluating all methods on physical metrics and perceptual scores. ZoneMaestro achieves 6.57 average, exceeding the best baseline by 1.29 points. Full results are in **Table 5** of the supplementary.
>
> **W4: Boundary-First Pipeline.**
> This is a thoughtful suggestion reflecting real-world workflows. We restructured the pipeline to accept a fixed floor plan first, then perform zone inventory and placement conditioned on this boundary. On SCALE, this variant yields an OOB of 0.11, Collision of 0.04, 23.12 assets, and overall score of 6.94. This is strongly on par with the standard pipeline score of 6.96, confirming the zone-graph factorization seamlessly adapts to floor-plan-first workflows.
>
> **W5: Figure Clarity.**
> We sincerely appreciate this feedback. We have prepared cleaner, re-annotated versions with reduced text density and larger fonts. The original Figure 2 is split into training pipeline and data pipeline views. Updated visuals are in **Figure A** of the supplementary.
>
> **KQ1: Multi-Reward GRPO Stability.**
> Training dynamics show staged convergence: boundary reward saturates around step 8, collision improves through step 25, zone disentanglement progresses gradually. No oscillation or policy collapse is observed. The alternating SFT–GRPO design is critical: pure RL tends toward sparse conservative layouts, and periodic SFT re-anchoring prevents mode collapse while retaining geometric gains.
>
> **KQ2: Reward Weight Ablation.**
> Due to the tight rebuttal timeline, we prioritized a leave-one-out ablation under the full alternating alignment setup ($\lambda_{\text{bound}}=1.0$, $\lambda_{\text{zone}}=0.5$, $\lambda_{\text{col}}=2.0$, KL $\beta=0.04$). Each reward targets a distinct failure mode: removing $R_{\text{bound}}$ spikes OOB from 0.09 to 0.15; removing $R_{\text{col}}$ raises collision from 0.04 to 0.11 with significant realism drop; removing $R_{\text{zone}}$ has the mildest effect since boundary and collision constraints already manage most physical violations. We hope this demonstrates each component's effectiveness. Full ablation metrics are in **Table 3** of the supplementary.

---

### Decision · Program_Chairs · 2026-04-30

**Decision:**

Accept (regular)

**Comment:**

The paper received 2 positive and 2 negative after rebuttal and discussion, in which reviewer CXTN mentioned that the concerns are fully addressed in the rebuttal thread but stated a bit differently in the final justification. Initially, reviewers had concerns about evaluations (e.g., metric, robustness of instruction-following, sensitivity analysis, intermediate reasoning correctness) and some technical clarity (e.g., small object handling, latency, training efficiency). After the rebuttal, except for reviewer 4rX4 who did not check the rebuttal, other reviewers are satisfied with the feedback. The AC took a closer look at the paper, reviews, and rebuttal, and found that the main concerns from reviewers are addressed well (including the one for reviewer 4rX4). Therefore, the AC recommends the acceptance rating, while highly encouraging the authors to improve the current version accordingly and release the dataset and code for reproducibility.